# One RL to See Them All: Visual Triple Unified Reinforcement Learning

## Abstract

Reinforcement learning (RL) has significantly advanced the reasoning capabilities of vision-language models (VLMs). However, its application beyond reasoning remains largely unexplored, especially for perception-intensive tasks like object detection and grounding. We propose **V-Triune**, a **V**isual **Tri**ple **U**nified Rei**n**forcement L**e**arning system that enables VLMs to jointly learn visual reasoning and perception tasks within a single training pipeline. V-Triune comprises three complementary components: *Sample-Level Data Formatting* to unify diverse inputs, *Verifier-Level Reward Computation* to deliver modular rewards via specialized verifiers, and *Source-Level Metric Monitoring* to enable fine-grained diagnostics. A key innovation within the verifier component is the proposed Dynamic IoU reward, which provides adaptive and progressive feedback for several perception tasks. Leveraging V-Triune, we develop Orsta (7B, 32B), a family of models built upon open-source backbones. Jointly training Orsta on a diverse dataset of eight representative reasoning (math, puzzle, etc.) and perception (detection, grounding, etc.) tasks leads to consistent improvements across both domains. As a result, Orsta achieves substantial gains on MEGA-Bench Core, with improvements ranging from +2.1 to +14.1 over its baselines, and these benefits extend to a wide range of downstream tasks. These results establish V-Triune as an effective and scalable system for building more comprehensive VLMs. Code is provided in the supplementary materials.

## 1 Introduction

While reinforcement learning (RL) has emerged as a powerful paradigm for post-training Vision Language Models (VLMs), the current research landscape remains fragmented (Liu et al., 2025d; Ma et al., 2025a; Tan et al., 2025; Liu et al., 2025c; Shen et al., 2025; Wang et al., 2025b). Prior works have largely specialized in one of two distinct domains: visual reasoning tasks like math and science QA (Huang et al., 2025; Yang et al., 2025; Meng et al., 2025; Wang et al., 2025a), where RL training mirrors established LLM paradigms, or a narrow subset of perception tasks like object detection (Yu et al., 2025a; Ma et al., 2025a; Liu et al., 2025b). This specialization has left a critical challenge largely unexplored: the creation of a unified RL framework that can jointly optimize a single VLM for both high-level reasoning and fine-grained perception.

This divide stems from a cascade of interlocking challenges that render existing RL frameworks ill-suited for unified VLM training. First, the heterogeneity of data requires managing diverse reward compositions even within a single task (e.g., some reasoning samples require only an accuracy reward, while others demand a joint accuracy-format reward with sample-specific weightings), demanding a level of fine-grained, sample-specific control that conventional systems lack. Second, this complexity is magnified by the fundamental incompatibility between reward paradigms: the exact-match verification required for reasoning tasks and the continuous spatial metrics (e.g., IoU) used in perception tasks like detection. Bridging these disparate paradigms in turn presents a major architectural hurdle, as a monolithic design would be brittle and unscalable. Finally, the joint training process itself is plagued by an opacity of dynamics, where aggregated metrics obscure source-specific failures, making it nearly impossible to diagnose instability or reward collapse.

To address these challenges, we introduce **V**isual **Tri**ple **U**nified Rei**n**forcement L**e**arning (V-Triune), a RL system to unify VLM training across reasoning and perception through a holistic

| Model | Size | Backbone | Ma | Sc | Ch | Pu | OCR | DET | GND | CNT | Others |
|---|---|---|---|---|---|---|---|---|---|---|---|
| Visual-RFT (Liu et al., 2025d) | 2B | Qwen2-VL | - | - | - | - | - | ✓ | ✓ | - | CLS |
| DeepPerception (Ma et al., 2025a) | 2B | Qwen2-VL Base | - | - | - | - | - | - | - | - | S.QA |
| Vision-R1 (Huang et al., 2025) | 7B,72B | Qwen2.5-VL | ✓ | ✓ | - | - | - | - | - | - | - |
| R1-Onevision (Yang et al., 2025) | 7B | Qwen2.5-VL | ✓ | ✓ | ✓ | - | - | - | - | - | V.QA |
| Reason-RFT (Tan et al., 2025) | 2B,7B | Qwen2-VL | ✓ | - | - | - | - | - | - | ✓ | S.QA |
| OThink-MR1 (Liu et al., 2025c) | 2B,7B | Qwen2.5-VL | ✓ | - | - | - | - | - | - | ✓ | - |
| Perception-R1 (Yu et al., 2025a) | 2B,3B | Qwen2 & 2.5-VL | - | - | - | - | ✓ | ✓ | ✓ | - | - |
| VLM-R1 (Shen et al., 2025) | 3B | Qwen2.5-VL | - | - | - | - | - | ✓ | ✓ | - | - |
| MM-EUREKA (Meng et al., 2025) | 7B,32B | Qwen2.5-VL | ✓ | - | - | - | - | - | - | - | S.QA |
| VL-Rethinker (Wang et al., 2025a) | 7B,32B,72B | Qwen2.5-VL | ✓ | ✓ | ✓ | - | - | - | - | - | S.QA |
| **Orsta (ours)** | 7B,32B | Qwen2.5-VL | ✓ | ✓ | ✓ | ✓ | ✓ | ✓ | ✓ | ✓ | - |

Table 1: Task-wise comparison with related works. The evaluated tasks include Math (Ma), Science (Sc), Chart (Ch), Puzzle (Pu), OCR, Detection (DET), Grounding (GND), Counting (CNT), Classification(CLS), Spatial question answering (S.QA), and Visual question answering (V.QA).

co-design of data formatting, reward computation, and training diagnostics. V-Triune systematically resolves the aforementioned conflicts through a tightly integrated, three-tier architecture: Sample-Level Data Formatting (§3.1) tackles the challenge of task diversity by allowing each data sample to self-define its own reward configuration and verifier, enabling unparalleled flexibility. Verifier-Level Reward Computation (§3.2) resolves the conflict between reward paradigms by delegating computation to specialized, modular verifiers. A key innovation within this framework is the Dynamic IoU Reward mechanism (§3.2.1), which gradually raises the localization precision bar as training progresses, guiding the model toward high-precision localization. Lastly, Source-Level Metric Monitoring (§3.3) ensures training stability and provides granular diagnostics in this complex multi-task setting by logging metrics on a per-source basis. Crucially, these components are not independent: sample-level declarations route data to the correct verifier, while source-level metrics inform adjustments to reward configurations, forming a closed-loop system for unified RL.

Leveraging the V-Triune system, we develop Orsta (**O**ne **R**L to **S**ee **T**hem **A**ll) model series (7B-32B). These models undergo joint optimization across a diverse set of eight tasks, spanning visual reasoning (mathematics, science, chart, puzzle) and visual perception (object detection, grounding, OCR, counting). On the comprehensive MEGA-Bench core (Chen et al., 2024) benchmark, which encompasses over 440 diverse, human-annotated tasks spanning reasoning, perception, and multimodal understanding, Orsta demonstrates substantial performance gains ranging from +2.1% to +14.1% across its variants compared to the base models. These benefits extend to prominent downstream benchmarks, including MMMU, MathVista, COCO, and CountBench, validating V-Triune's effectiveness and scalability. Our core contributions are: 1) We introduce V-Triune, an unified, scalable, and extensible RL system for jointly training VLMs on both visual reasoning and perception tasks. 2) We propose the Dynamic IoU reward, a novel, adaptive reward mechanism that significantly enhances stability and performance for visual perception tasks like detection and grounding. 3) We establish a comprehensive training recipe, with key engineering optimizations, for stable and effective RL training across eight diverse tasks. 4) We present Orsta, a family of high-performance models trained with V-Triune, achieving strong performance across various benchmarks.

## 2 RELATED WORK

Recent advancements in visual RL (summarized in Tab. 1) have explored diverse strategies to enhance multimodal reasoning and perception. However, these efforts have largely bifurcated into two specialized tracks. One major line of work has focused on enhancing visual reasoning. Models such as Vision-R1 (Huang et al., 2025), LMM-R1 (Peng et al., 2025b), R1-OneVision (Yang et al., 2025), VisualThinker-R1-Zero (Zhou et al., 2025), MM-Eureka (Meng et al., 2025), and PeBR (Chen et al., 2025b) primarily leverage reasoning-heavy datasets and rule-based signals to elicit complex problem-solving capabilities, closely mirroring RL paradigms from the LLM domain.

In parallel, another stream of research has applied RL to improve perception-intensive tasks. Models like Visual-RFT (Liu et al., 2025d), R1-V (Chen et al., 2025a), Reason-RFT (Tan et al., 2025), and DeepPerception (Ma et al., 2025a) apply task-specific, verifiable reward signals for tasks like detec-

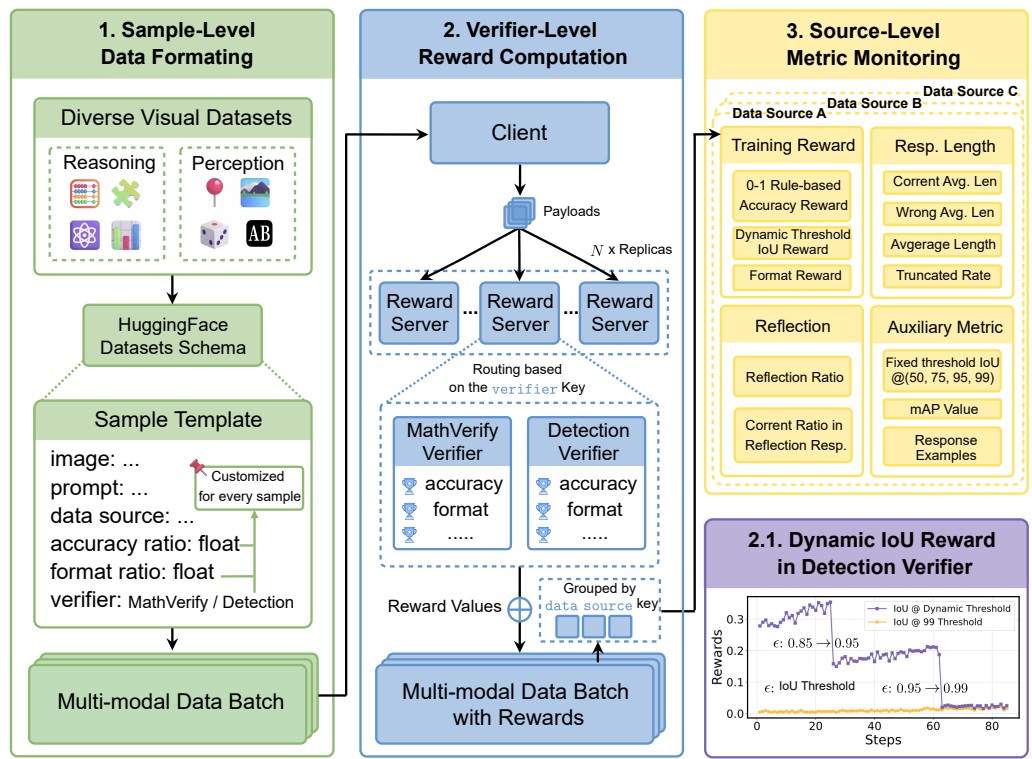

Figure 1: **The V-Triune System**. It integrates three core components operating at distinct levels: 1) Sample-Level Data Formatting to unify diverse task inputs; 2) Verifier-Level Reward Computation for custom rewards, which incorporates our novel Dynamic IoU reward for perception tasks like detection; and 3) Source-Level Metric Monitoring to diagnose problems at the data-source level.

tion, grounding, and counting. Others, such as Seg-Zero (Liu et al., 2025a), Perception-R1 (Yu et al., 2025a), and VLM-R1 (Shen et al., 2025), further propose tailored reward functions for segmentation and OCR, though they often remain within task-specific generalization boundaries.

Several approaches have also explored broader cross-task generalization. OThink-MR1 (Liu et al., 2025c) incorporates GRPO with Dynamic KL, while VL-Rethinker (Wang et al., 2025b) introduces selective sample replay and forced rethinking to enhance reasoning. VisionReasoner (Liu et al., 2025b) reformulates diverse perception sub-tasks into a multi-goal problem, and Vision-G1 (Zha et al., 2025) introduces data curation strategies for diverse visual reasoning tasks. In contrast to these prior efforts that largely treat reasoning and perception in isolation, our proposed V-Triune unifies both domains under a single, comprehensive RL system.

## 3 V-TRIUNE: VISUAL TRIPLE UNIFIED REINFORCEMENT LEARNING

This section describes V-Triune, our visual triple unified RL system. As shown in Fig. 1, V-Triune jointly trains VLMs on diverse tasks spanning visual reasoning and perception within a unified pipeline. The system comprises three core, interconnected components designed to collectively handle these diverse tasks. The following subsections detail these three components, highlighting our novel Dynamic IoU reward mechanism.

### 3.1 SAMPLE-LEVEL DATA FORMATTING

This section introduces how data is formatted to support unified training across perception and reasoning tasks. A key challenge is accommodating the diverse reward computations, components, and weighting strategies required by different tasks. For example, tasks like math, puzzle, and OCR compute rewards based on the correctness of textual answers, while detection and grounding tasks

rely on spatial metrics such as IoU and bounding box formatting. In conventional RL setups, reward computation is typically defined at the task level. While this allows modular reward functions to be implemented externally, it limits flexibility when fine-grained control is required. Many tasks may contain heterogeneous samples that demand different reward strategies. For instance, some samples require only an accuracy-based reward, while others demand a joint reward that accounts for accuracy and format, with sample-specific weightings, necessitating per-sample reward configurations.

To achieve this flexibility, we define reward configurations at sample level. Each sample specifies its reward types, relative weights, and associated verifier, enabling automatic reward routing and fine-grained weighting without modifying the core training logic.

Our data schema (the left of Fig. 1), implemented with HuggingFace Datasets, extends common fields like `images` and `prompt` with several task-agnostic keys for reward control. Specifically, we include: 1) `accuracy_ratio` and `format_ratio` to enable, disable or reweight different types of rewards on a per-sample basis by adjusting these ratios; 2) a `verifier` field to specify which verifier should be used for reward computation. (detailed in Sec. 3.2); and 3) `data_source` field to indicate sample origin and enable source-level metric monitoring (Sec. 3.3). Fig. 9 illustrates the complete data format, a design that reinforces the aforementioned sample-level flexibility. In summary, this sample-level design is the key to integrating diverse datasets into a unified pipeline while maintaining flexible, scalable reward control.

## 3.2 VERIFIER-LEVEL REWARD COMPUTATION

A core design principle of V-Triune is to handle diverse tasks through a modular and scalable reward computation mechanism. To achieve this, we introduce the *verifier-level*, a computation granularity where monolithic reward functions are replaced by specialized verifiers for specific task groups. To implement this principle, we designed a standalone, asynchronous client-server architecture, illustrated in the middle of Fig. 1. Decoupling reward computation from the main training loop enhances modularity and scalability. In this architecture, requests are routed to the appropriate verifier based on the `verifier` field in each sample's metadata, ensuring flexibility and extensibility. To support the diverse visual reasoning and perception tasks involved in this work, we implement two primary verifiers, each encapsulating distinct, rule-based reward logic:

**MathVerifyVerifier: For Reasoning, Counting, and OCR**  This verifier handles tasks with short-form textual answers that can be parsed and unambiguously verified by deterministic rules, such as reasoning, counting, and OCR. To mitigate reward hacking (Weng, 2024), it employs a simple yet robust binary (0-1) accuracy-based reward function:

$$R_{\text{acc}}(\hat{a}, a) = \mathbb{I}(\texttt{verify}(\texttt{parse}(\hat{a}), \texttt{parse}(a))) \tag{1}$$

where the predicted answer $\hat{a}$ is parsed from model output, (instructed to be enclosed in `\boxed{}`, and verified against the ground-truth $a$ using `math_verify` (Kydlíček, 2025).

**DetectionVerifier: For Detection and Grounding**  This verifier manages tasks that require spatial understanding and produce structured outputs, such as COCO-style `JSON`. Initial experiments revealed that the model struggled with the `\boxed` format in JSON but readily learned to use `<answer>` tags. Consequently, this verifier computes a composite reward based on both format and accuracy. First, to enforce the correct output structure, a format reward is defined as:

$$R_{\text{format}}(o_q) = 0.25 \sum_{i=1}^{4} \mathbb{I}(\text{count}(o_q, s_i) = 1) \tag{2}$$

where $o_q$ represents the model's response to question $q$, and $s_i$ denotes a specific format tag ($\{s_i\}_{i=1}^{4} = \{\texttt{<think>},\texttt{</think>},\texttt{<answer>},\texttt{</answer>}\}$). The indicator function $\mathbb{I}(\text{condition})$ evaluates to 1 if the condition is true, and 0 otherwise.

For the accuracy component, we adapt the standard Intersection over Union (IoU) (Everingham et al., 2010) metric, a common practice in object detection and adopted in prior works (Liu et al., 2025d; Yu et al., 2025a; Shen et al., 2025), with formulations provided as:

$$R_{\text{acc}}(\hat{a}, a) = \begin{cases} \text{IoU}(\hat{a}, a), & \text{if } \text{IoU}(\hat{a}, a) \geq \epsilon \\ 0, & \text{else} \end{cases}, \quad \text{where} \quad \text{IoU}(\hat{a}, a) = \frac{\text{Area}(\hat{a} \cap a)}{\text{Area}(\hat{a} \cup a)} \tag{3}$$

where $\hat{a}$ represents the predicted bounding box, and $a$ denotes the golden bounding box. The threshold $\epsilon$ controls the strictness of the reward function, with higher values enforcing tighter matches. We ultimately adopt a dynamic $\epsilon$ finally, as detailed in Sec. 3.2.1. Finally, the overall reward, combining accuracy and format, is defined as: $\alpha_{\text{acc}} \cdot R_{\text{acc}} + \alpha_{\text{format}} \cdot R_{\text{format}}$, where $\alpha_{\text{acc}}$ and $\alpha_{\text{format}}$ are sample-specific weighting coefficients specified in the data format.

This verifier-level approach grants significant flexibility, modularity, and extensibility. For instance, supporting a new task group merely requires implementing a new verifier, leaving the core training pipeline untouched. Finally, the computed rewards are returned and integrated into their corresponding data samples for subsequent training and metric logging.

### 3.2.1 DYNAMIC IoU REWARD IN DETECTIONVERIFIER

For detection and grounding tasks, a robust reward signal is crucial for guiding the model toward fine-grained localization accuracy. We adopt a reward strategy based on IoU, as it provides a direct, interpretable, and controllable signal that is strongly aligned with task objective. While an IoU-based reward is a direct and robust choice, selecting a fixed threshold presents a fundamental dilemma.

On one hand, relaxed thresholds like IoU@50 (a common choice in Liu et al. (2025d); Yu et al. (2025a)) can be too lenient as an RL reward signal. This leniency, especially when contrasted with the strict exact-match rewards in reasoning tasks, creates reward ambiguity. It allows multiple, distinct predicted bounding boxes to receive the same reward, and this ambiguous signal fails to provide a clear gradient for the model to refine its localization precision. On the other hand, a potential solution to this ambiguity is to enforce a highly stringent threshold, such as $\epsilon = 0.99$. This approach enhances consistency between perception and reasoning signals and provides an unambiguous target. However, its extreme stringency leads to a severe reward sparsity problem, as most initial predictions receive zero reward, hindering learning.

To resolve this dilemma, we adopt a dynamic IoU reward strategy. The IoU threshold is progressively tightened: starting at 0.85 for the initial 10% of training, increasing to 0.95 until the 25% mark, and finally settling at 0.99 for the remainder. This staged approach results in a distinctive step-like reward pattern, where the training reward drops with each increase in difficulty and subsequently recovers as the model adapts, as demonstrated in the lower right of Fig. 1. This curriculum-based approach thus resolves the trade-off between reward ambiguity and sparsity.

### 3.3 SOURCE-LEVEL METRIC MONITORING

Monitoring training metrics is essential for understanding model dynamics and real-time issue diagnosis. However, for multi-task, multi-source training, aggregated or single-task metrics are often insufficient due to lack of traceability and per-source data variations. Therefore, we adopt source-level metric monitoring, which involves logging metrics per `data_source` for each batch. This approach helps identify faulty data sources, enables targeted debugging, and reveals cross-source learning interactions. This granular monitoring is particularly vital in RL, where training instability requires the more granular diagnostic capabilities that our source-level detail provides over standard built-in logging. This design has been instrumental in diagnosing and resolving numerous training issues, which are detailed in Appx. B as a contribution to the community.

As shown in the upper right of Fig. 1, data batches are grouped by data source for metric logging after rewards are computed. We designed a comprehensive set of metrics, organized into four categories, all of which are logged on a per-source basis. We log a detailed breakdown of *Training Reward*, including rule-based accuracy, output format, and IoU-based signals, to directly assess how well the model is learning specific data source. In parallel, we track *Response Length* metrics, such as average lengths and truncation rates, to analyze the model's generative patterns and diagnose critical issues like undesirable verbosity or potential model collapse.

For deeper insights and overall stability, we monitor two additional metric suites. To understand the model's reasoning process, we monitor *Reflection* metrics inspired by Ma et al. (2025b), computing the reflection ratio by tracking a curated list of 15 reflective words from their work, alongside the correctness rate within these reflection responses (see Appx. H for details). This lightweight analysis helps distinguish productive self-correction from inefficient "overthinking". Finally, a suite of *Auxiliary Metrics* provides a comprehensive overview of secondary objectives and training stability.

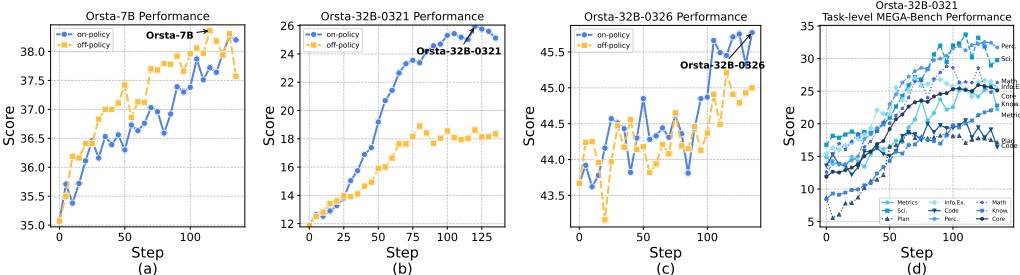

Figure 2: Training trends of on-policy vs off-policy across three model variants on MEGA-Bench core: 7B-(a), 32B-0321-(b) 32B-0326-(c) and task level performance of Orsta-32B-0321-(d). Models are evaluated every 5 generation steps from 0 to 135. The dark line in (d) denotes the task-level MEGA-Bench Core, linking to the performance shown in the blue line of (b).

These include standard perception evaluations, such as mAP and IoU at fixed thresholds (@50, 75, 95, 99), complemented by logging of sample responses for qualitative assessment of model behavior.

# 4 EXPERIMENT

## 4.1 IMPLEMENTATION DETAILS

**Models and Data:** We adopt Qwen2.5-VL-7B-Instruct and Qwen2.5-VL-32B-Instruct (0321 and 0326 versions, denoting their release dates) as our base models, chosen for their strong performance across both vision-language reasoning and perception. We curated a multi-task dataset of 47.7K high-quality samples from 18 sources, covering four reasoning and four perception tasks. Detailed information on data sources, filtering, and hyperparameters is provided in Appx. A.

**Training Recipe:** V-Triune is implemented upon verl (Sheng et al., 2024). All experiments are conducted on 64 NVIDIA H20 GPUs. Informed by our source-level monitoring, we implemented several key strategies to ensure stable and scalable training: freeze the ViT to prevent gradient explosion, filter leaked image tokens in responses, utilize a prompt pool to mitigate variance, and optimize system memory management. A detailed analysis of these strategies is provided in Appx. B.

**Experimental Settings:** We employ a modified version of Group Relative Policy Optimization (GRPO) (Shao et al., 2024) for all experiments. Inspired by recent work (Yu et al., 2025b; Hu et al., 2025), we remove the reference model and its associated KL loss to improve exploration and training efficiency (Hu et al., 2025; Schulman, 2020), and apply the clip-high trick with token-level loss to further enhance stability (Yu et al., 2025b). We compare two configurations: a standard on-policy setup and an off-policy setup with 8 optimization steps per rollout. Both are trained for 3 epochs with a rollout batch size of 1024, generating 8 candidate sequences per prompt. We freeze the ViT and connector modules for stability. Learning rates are $1 \times 10^{-6}$ (on-policy) and $5 \times 10^{-7}$ (off-policy) with a 5% warmup. Rollouts use the vLLM engine with sampling (temperature=1.0, top-p=1.0, max length=2048), while testing uses greedy decoding.

**Evaluation Benchmarks:** To comprehensively assess our model's capabilities, we evaluate its performance across three key domains: real-world tasks, visual reasoning, and visual perception. For a holistic evaluation on real-world tasks, we employ the core subset of MEGA-Bench (Chen et al., 2024) which consists of 440 diverse, long-tailed tasks encompassing over 6,000 expert-curated samples. For visual reasoning, we adopt MMMU (Yue et al., 2024) and MathVista (Lu et al., 2023). For visual perception, our evaluation suite includes COCO (Lin et al., 2014), OVDEval (Yao et al., 2023), CountBench (Paiss et al., 2023), OCRBench (Fu et al., 2024), and ScreenSpot-Pro (Li et al., 2025). Detailed evaluation protocols for each benchmark, including specific metrics, scoring methods, and implementation details, are provided in Appx. D.

Table 2: Performance comparison on MEGA-Bench core. Models with improved reasoning are marked by ♀; final scores are reported as weighted averages. QwenVL-2.5-32B-0321 is a version with incomplete post training, which are resolved in the 0326 version. All results are obtained using the official MEGA-Bench evaluation code, except for Gemma3-27B (†).

| Model | Knowledge | Mathematics | Perception | Coding | Info. Ex. | Planning | Science | Metrics | MEGA-Bench Core |
|---|---|---|---|---|---|---|---|---|---|
| 7B+ Model | | | | | | | | | |
| QwenVL-2-7B (Wang et al., 2024) | 39.96 | 25.95 | 39.99 | 31.49 | 40.29 | 16.64 | 28.59 | 43.61 | 34.47 |
| QwenVL-2.5-7B (Bai et al., 2025) | 38.84 | 27.67 | 41.24 | 28.93 | 50.23 | 16.32 | 36.75 | 41.64 | 35.07 |
| InternVL-3-8B (Zhu et al., 2025) | 36.64 | **32.75** | 42.17 | **35.11** | 48.92 | 14.35 | 36.51 | **53.94** | 36.48 |
| Gemma3-12B (Gemma et al., 2025) | 41.11 | 29.10 | 37.38 | 30.27 | 46.56 | 16.10 | 36.83 | 50.40 | 35.04 |
| Kimi-VL-A3B (Kimi et al., 2025) | 37.63 | 27.07 | 39.50 | 22.30 | 40.99 | **22.17** | 33.94 | 46.65 | 34.40 |
| MM-Eureka-7B ♀ (Meng et al., 2025) | 40.12 | 31.59 | 39.71 | 28.75 | 49.32 | 16.64 | 37.25 | 46.39 | 35.96 |
| VL-Rethinker-7B ♀ (Wang et al., 2025a) | 40.65 | 30.08 | 42.02 | 29.87 | 52.03 | 17.83 | 36.82 | 46.90 | 37.25 |
| Kimi-VL-A3B-Thinking ♀ | 33.45 | 17.76 | 28.11 | 14.69 | 41.14 | 12.64 | 28.60 | 43.97 | 27.08 |
| **Orsta-7B (Ours)** ♀ | **41.65** | 31.48 | **43.84** | 32.82 | **54.07** | 17.83 | **36.91** | 41.66 | **38.31** |
| △ (Ours - Backbone) | +2.8 | +3.8 | +2.6 | +3.9 | +3.8 | +1.5 | +0.2 | +0.0 | +3.2 |
| 32B+ Model | | | | | | | | | |
| QwenVL-2.5-32B-0321 (Bai et al., 2025) | 8.48 | 12.62 | 11.99 | 13.59 | 15.44 | 8.61 | 16.78 | 14.91 | 11.87 |
| MM-Eureka-32B ♀ (Meng et al., 2025) | 12.20 | 20.19 | 21.88 | 15.86 | 21.23 | 15.47 | 19.95 | 22.77 | 18.57 |
| VL-Rethinker-32B ♀ (Wang et al., 2025a) | 12.16 | 28.09 | 22.99 | 11.89 | 21.50 | 15.09 | 28.10 | 15.73 | 19.41 |
| **Orsta-32B-0321 (Ours)** ♀ | **21.33** | **28.55** | **32.23** | **19.44** | **26.38** | **17.78** | **33.20** | **24.18** | **25.94** |
| △ (Ours - Backbone) | +12.9 | +15.9 | +20.2 | +5.9 | +10.9 | +9.2 | +16.4 | +9.3 | +14.1 |
| Gemma3-27B † (Gemma et al., 2025) | 49.43 | 42.20 | 45.46 | 40.18 | 49.30 | 24.96 | 47.08 | 58.99 | 41.82 † |
| QwenVL-2.5-32B-0326 (Bai et al., 2025) | 46.09 | 32.04 | 47.55 | 38.36 | 61.65 | 28.43 | 37.55 | 50.38 | 43.67 |
| InternVL-3-38B (Zhu et al., 2025) | 46.32 | **40.29** | **55.05** | **45.29** | 56.63 | 22.88 | **52.04** | **58.04** | **46.69** |
| Skywork-R1V-38B ♀ (Peng et al., 2025a) | 25.59 | 28.45 | 22.95 | 19.88 | 19.53 | 9.74 | 22.64 | 37.55 | 21.54 |
| Skywork-R1V2-38B ♀ (Peng et al., 2025a) | 17.08 | 12.38 | 15.65 | 7.14 | 9.90 | 17.60 | 14.29 | 0.0 | 15.39 |
| **Orsta-32B-0326 (Ours)** ♀ | **46.78** | 37.43 | 50.86 | 38.92 | **63.14** | 28.05 | 42.68 | 53.01 | 45.77 |
| △ (Ours - Backbone) | +0.7 | +5.4 | +3.3 | +0.6 | +1.5 | -0.4 | +5.1 | +2.6 | +2.1 |

## 4.2 Performance and Analysis

### 4.2.1 MEGA-Bench

Tab. 2 compares Orsta against its backbone and other leading general-purpose/reasoning-enhanced VLMs. Orsta shows consistent gains at both 7B and 32B scales: Orsta-7B achieves 38.31 (+3.2) on MEGA-Bench Core, and Orsta-32B-0326 reaches 45.78 (+2.1). V-Triune notably boosts performance in domains with enriched training data—mathematics (+3.8 at 7B, +5.4 at 32B), perception, planning, and science—indicating strong generalization in both reasoning and perception tasks. More impressively, our method's benefits generalize to out-of-domain tasks, with coding and metrics showing performance gains despite the absence of corresponding training data.

Beyond these top-line numbers, a deeper analysis of the two 32B variants reveals the core impact of our V-Triune system. The Orsta-32B-0321 variant serves as an ideal case study for the alignment power of our approach. Its underlying checkpoint, Qwen2.5-VL-0321, is a publicly released version with known deficiencies in perception and output formatting, a finding confirmed by both our evaluations and prior work like Wang et al. (2025a). These issues are addressed in the subsequent 0326 release. We therefore consider 0321 version a strong baseline with significant untapped potential that our RL framework aims to unlock. V-Triune successfully unlocks this potential: as shown in Tab. 2 and Fig. 2(d), in-domain tasks covered by our training data see substantial gains exceeding 10%. More significantly, out-of-domain tasks like coding and metrics also achieve a 5-10% performance boost. This demonstrates that our unified RL training acts as a powerful, general-purpose alignment mechanism, enhancing the model's overall capabilities through strong generalization. Furthermore, the results from the already well-aligned 32B-0326 variant confirm that V-Triune is also a potent

Table 3: Performance on common downstream tasks. We report the mAP|mAP@50 for COCO, NMS-mAP for OVDEval and accuracy for others. We compute mAP score on COCO using the standard cocoapi, with full evaluation details in Appx. E.

| Tasks | QwenVL-2.5-7B | Orsta-7B | QwenVL-2.5-32B 0321 | Orsta-32B 0321 | QwenVL-2.5-32B 0326 | Orsta-32B 0326 |
|---|---|---|---|---|---|---|
| Visual Reasoning | | | | | | |
| MMMU$_{val, rule-score}$ (Yue et al., 2024) | 45.56 | 49.70 | 37.11 | 34.67 | 39.22 | 38.00 |
| MMMU$_{val, gpt-score}$ (Yue et al., 2024) | 54.40 | 57.10 | 60.80 | 64.11 | 64.20 | 64.78 |
| MathVista$_{testmini}$ (Lu et al., 2023) | 67.50 | 72.50 | 70.80 | 76.30 | 73.40 | 76.40 |
| Visual Perception | | | | | | |
| COCO$_{val, single-object}$ (Lin et al., 2014) | 79.20 \| 89.79 | 80.73 \| 91.45 | 2.09 \| 2.33 | 52.99 \| 62.93 | 79.30 \| 94.19 | 79.10 \| 93.76 |
| COCO$_{val, multiple-object}$ (Lin et al., 2014) | 32.12 \| 42.65 | 41.41 \| 56.06 | 0.88 \| 1.01 | 28.65 \| 38.35 | 42.16 \| 56.39 | 45.43 \| 60.98 |
| COCO$_{val, full}$ (Lin et al., 2014) | 33.63 \| 44.16 | 42.45 \| 56.92 | 0.88 \| 1.00 | 29.29 \| 38.89 | 43.28 \| 57.43 | 46.39 \| 61.81 |
| OVDEval (Yao et al., 2023) | 52.23 | 57.52 | 3.84 | 35.65 | 56.80 | 60.36 |
| ScreenSpot-Pro (Li et al., 2025) | 22.71 | 23.91 | 49.46 | 50.98 | 51.23 | 52.69 |
| CountBench (Paiss et al., 2023) | 71.69 | 82.28 | 83.71 | 88.59 | 83.91 | 88.19 |
| OCRBenchv2 (Fu et al., 2024) | 55.11 | 56.05 | 44.85 | 43.78 | 58.14 | 59.09 |

fine-tuning tool. Despite this model's strong initial performance, our framework still elicits considerable gains, pushing its capabilities even further.

In addition to the final scores, the training dynamics themselves offer valuable insights into the optimization behavior of different RL strategies. As shown in Fig. 2, the MEGA-Bench learning curves reveal that the relative effectiveness of on-policy versus off-policy RL varies across model sizes. For the 7B variant, off-policy training leads to faster convergence and higher final performance, whereas for the larger 32B models (32B-0321 and 32B-0326), on-policy training, despite its slower initial progress, achieves more sustained improvement and higher peak performance. These results indicate that neither strategy consistently outperforms the other; instead, their relative merits appear to differ depending on the specific model. In practice, it is advisable to evaluate both approaches empirically and select the one that yields better performance for the given model and task.

In summary, our results show that V-Triune is a versatile system: it can both unlock the latent potential of less-aligned models and further polish state-of-the-art ones, solidifying the role of unified RL as a powerful and general-purpose alignment strategy for VLMs.

### 4.2.2 COMMON DOWNSTREAM TASKS

As shown in Tab. 3, MMMU shows consistent gains on the more reliable gpt-score evaluation, with improvements of +2.7% (7B), +3.3% (32B-0321), and +0.6% (32B-0326). On MathVista, it achieves around 5% gains. These results align with improvements seen on math tasks in MEGA-Bench, reinforcing Orsta's strength in enhancing reasoning capabilities.

Orsta models demonstrate consistent improvements in visual perception on the COCO benchmarks. Specifically, Orsta-7B delivers notable gains, particularly in the more challenging multi-object detection task with a +9.29 mAP increase. Orsta-32B-0321 dramatically rectifies the perception shortcomings of its baseline, achieving massive boosts of over 50 mAP for single-object and nearly 28 mAP for multi-object detection. Building on a strong foundation, Orsta-32B-0326 further advances multi-object performance by +3.27 mAP, while maintaining a similar level of performance on single-object detection. On OVDEval, Orsta-7B and 32B-0326 improve by +5.3 and +3.5 mAP. The trend of broad improvements continues on other benchmarks: while GUI and OCR tasks see modest but consistent gains, CountBench shows the most remarkable boost, with Orsta-7B approaching the performance of 32B baselines and Orsta-32B. Overall, V-Triune delivers greater perception improvements for less-aligned base models (0321) than for well-aligned ones (0326).

### 4.2.3 TRAINING DYNAMICS ANALYSIS

We conducted an in-depth analysis of the model's training dynamics using V-Triune's source-level metric monitoring. As shown in Fig. 7, the results reveal distinct behavioral patterns across different

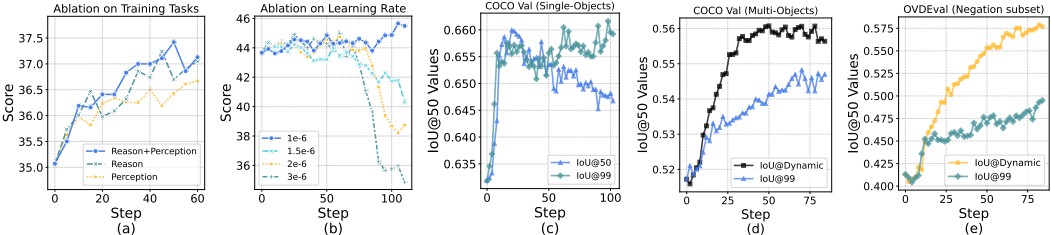

Figure 3: Ablation studies. (a) Different task compositions evaluated on the 7B model; (b) Effects of learning rates on the performance of the 32B model. (a-b) show Megabench-core performance over generation steps. (c-e) Effectiveness of the dynamic IoU reward strategy on the 7B model.

task categories. For reasoning-intensive tasks like Math and Puzzle-solving, the model progressively increases its response length, suggesting it learns to employ longer, more elaborate chains of thought. In stark contrast, for the perception task of Detection, the model exhibits an opposite trend: its responses become more concise. This suggests that for tasks requiring precise, direct outputs, the model learns to suppress verbose reasoning in favor of more efficient, direct prediction. Interestingly, the OCR task presents a hybrid pattern. While also a perception task, its response length show a upward trend, similar to the reasoning tasks. This divergence within perception tasks strongly supports a hypothesis that the model activates distinct cognitive pathways tailored to specific task demands, rather than adopting a uniform strategy. A detailed analysis is provided in Appx. C.

### 4.3 ABLATION STUDY

**Task-composition ablation** As shown in Fig. 3a, training on both reasoning and perception data yields the strongest performance, reaching approximately 37.5. The reasoning-only data performs nearly as well but the peak performance is slightly lower, which suggests that the benchmark may prioritize logical competence over pure perception. Perception-only data consistently lags by 0.5–1.0 points but still shows steady improvement. The consistent performance hierarchy (Reasoning+Perception > Reasoning > Perception) underscores the value of mixed-task corpora: combining complementary signals leads to additive gains rather than diluted optimization.

**Learning-rate ablation** For the Orsta-32B-0326 model (Fig. 3b), a learning rate of 1e-6 yields the highest and most stable plateau (45.5), while 1.5e-6 performs similarly until a mild degradation after 80 steps. Increasing the rate to 2e-6 causes a late-stage collapse to around 38, and 3e-6 diverges catastrophically after 50 steps, dropping below 36. This pattern suggests that larger models sit closer to the edge of the loss landscape, benefiting from small, stable updates, with 1e-6 offering the best trade-off between convergence speed and final performance on MEGABench.

**Reward strategy ablation** Our ablation study on the reward strategy, conducted on a data subset containing only detection and grounding tasks, validates our dynamic IoU mechanism by revealing the clear limitations of fixed-threshold approaches. Fig. 3c, which evaluates fixed thresholds, highlights their inherent trade-off: a lenient IoU@50 reward proves unstable and leads to performance collapse in later stages, while a strict IoU@99 reward is stable but learns inefficiently. In contrast, subplots Fig. 3d and Fig. 3e demonstrate that our dynamic IoU reward consistently and significantly outperforms the strict IoU@99 baseline on the more challenging COCO multi-object and OVDEval negation subsets, proving its superior performance and learning efficiency.

**Extended Reward strategy ablation** We further justify our Dynamic IoU design ($0.85 \rightarrow 0.95 \rightarrow 0.99$) via more rigorous analysis (see Fig. 11 & Appx. J). Comparisons with ""Loose" ($0.7 \rightarrow 0.8 \rightarrow 0.85$) and "Strict" ($0.9 \rightarrow 0.95 \rightarrow 0.99$) variants demonstrate that our schedule optimally balances early-stage learnability (targeting an initial reward in a moderate range, e.g. 0.2-0.4) with late-stage precision enforcement (eliminating reward ambiguity), serving as a robust strategy aligned with the model's intrinsic learning trajectory.

### 5 DISCUSSION & FUTURE WORK

We introduced V-Triune, a unified reinforcement learning system that successfully bridges the gap between reasoning and perception tasks in VLMs. Through its modular three-tier design and a

novel dynamic IoU reward, we demonstrate substantial performance gains across a broad spectrum of benchmarks, establishing a new, scalable paradigm for VLM post-training.

While our results are promising, we also highlight key avenues for future research. These include establishing clearer performance scaling laws for perception-intensive tasks and further exploring RL to overcome the data bottlenecks of supervised fine-tuning. We believe the continued exploration of unified RL holds the key to building more general and capable vision-language models.

## REPRODUCIBILITY STATEMENT

Full details necessary for reproducibility are provided in the paper and supplementary materials. The V-Triune system architecture is described in Sec. 3, with the Dynamic IoU reward detailed in Sec. 3.2.1. Implementation specifics—including model backbones (Qwen2.5-VL-7B/32B), training data composition (47.7K samples across 8 tasks), and hyperparameters—are given in Sec. 4.1 and Appx. A. Engineering optimizations are documented in Appx. B. Evaluation protocols for MEGA-Bench, COCO, and other benchmarks are specified in Sec. 4.1 and Appx. D and E. The complete data schema is illustrated in Fig. 4 and Appx. G. Anonymized code, model checkpoints, training scripts, and the curated dataset are included in the supplementary materials.

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

## LLM USAGE

The authors used a LLM solely for minor grammatical refinement and language polishing of the manuscript. The LLM did not contribute to the conception, design, analysis, interpretation of the research, nor to the generation of any novel ideas, arguments, or content. All scientific content, methodology, results, and conclusions were developed exclusively by the human authors.

# A  DATA CURATION

Table 4: Data source composition and curation. The curated corpus contains 27,133 reasoning samples (Math, Puzzle, Science, Chart; 56.8%) and 20,633 perception samples (Detection, Grounding, Counting, OCR; 43.2%), for a total of 47,766 examples (47.7K).

| Task | Count (Proportion) | Data source name | After curation | Original count | Notes |
|---|---|---|---|---|---|
| Math | 11,810 (24.72%) | mm math | 3,539 | 5,901 | |
| | | geometry3k | 2,539 | 3,002 | |
| | | mmk12 | 5,732 | 15,616 | |
| Puzzle | 5,980 (12.52%) | PuzzleVQA + AlgoPuzzleVQA | $2,648 \times 2$ | $3,800 \times 2$ | Puzzle data are duplicated because the original puzzle data size is relatively small. |
| | | VisualPuzzles | $342 \times 2$ | $1,168 \times 2$ | Puzzle data are duplicated because the original puzzle data size is relatively small. |
| Science | 4,339 (9.08%) | ScienceQA | 536 | 4,114 | |
| | | SciVQA | 1,264 | 15,120 | |
| | | ViRL39K ("STEM" & "Science") | 2,539 | 4,431 | |
| Chart | 5,004 (10.48%) | ChartQAPro | 498 | 1,948 | |
| | | ChartX | 2,353 | 4,848 | |
| | | Table-VQA-Bench | 496 | 1,500 | |
| | | ViRL39K (Tables / Diagrams / Charts) | 1,657 | 6,189 | |
| Detection | 8,000 (16.75%) | V3Det | 4,000 | 15,000 | We randomly sample a 15k subset from 183,354 images; after filtering we obtain 6,287 samples and then randomly select 4k. |
| | | Object365 | 4,000 | 15,000 | We randomly sample a 15k subset from 1.74M images; after filtering we obtain 8,889 samples and then randomly select 4k. |
| Grounding | 4,870 (10.20%) | $D^3$ | 4,870 | 20,278 | |
| Count | 1,725 (3.61%) | CLEVR | 1,725 | 4,000 | We sample a 4k subset from the full 35k dataset. |
| OCR | 6,038 (12.64%) | LLaVA-OneVision (OCR-en) | 3,092 | 8,000 | We sample 8k images from the 56,613 images in the ocr_vqa category of LLaVA-OneVision-Mid-Data. |
| | | EST-VQA | 2,946 | 8,000 | We sample 7k images from the 17,047 images in the EST-VQA training set. |

We select four reasoning tasks—Math, Puzzle, Science, and Chart—for their varied reasoning demands, and four perception tasks—Detection, Grounding, Counting, and OCR—for their broad coverage of visual understanding. Data sources for each task are listed below:

- For the Math task, mm_math (Sun et al., 2024), geometry3k (Lu et al., 2021), and mmk12 (Meng et al., 2025) are chosen.

- For the Puzzle task, PuzzleVQA(Chia et al., 2024) and AlgoPuzzleVQA(Ghosal et al., 2024) are merged due to their shared origin, and VisualPuzzles (Song et al., 2025) is additionally included.

- For the Science task, ScienceQA (Lu et al., 2022), SciVQA (Borisova & Rehm, 2025), and the "Broader STEM Topics" and "(GradeSchool) Science" categories from ViRL39K (Wang et al., 2025b) are used.

- For the Chart task, ChartQAPro (Masry et al., 2025), ChartX (Xia et al., 2024), Table-VQA (Kim et al., 2024), and the Tables/Diagrams/Charts categories from ViRL39K (Wang et al., 2025b) are used.

- For the Detection task, V3Det (Xie et al., 2023) and Object365 (Shao et al., 2019) are chosen.

- For the Grounding task, $D^3$ (Xie et al., 2023) is used.

- For the Counting task, CLEVR (Johnson et al., 2017; Tan et al., 2025) is used.

- For the OCR task, English OCR questions are extracted from LLaVA-OV Data (Li et al., 2024) and EST-VQA (Wang et al., 2020).

To reduce noise, we apply a two-stage data filtering process (Figure Fig. 4): (1) rule-based filtering and (2) difficulty-based filtering. This yields 47.7K high-quality samples across 18 datasets and 8 tasks. To mitigate dataset bias, puzzle data is duplicated to ensure sufficient coverage. The final corpus includes approximately **20.6K perception** and **27.1K reasoning** samples, primarily consisting of single-image, single-turn conversations.

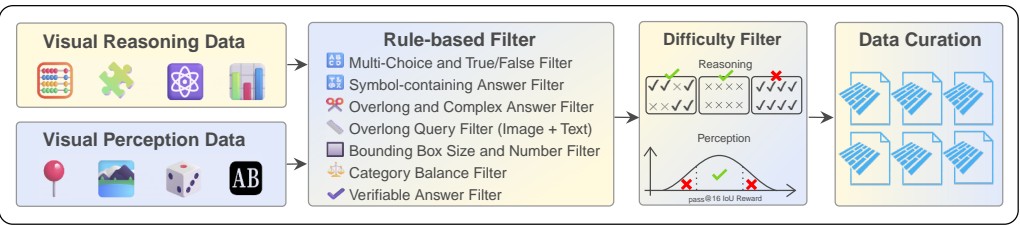

Figure 4: **Data Curation Process**. First, visual reasoning and visual perception data pass through a rule-based filter, which removes samples that do not meet preset criteria. Subsequently, the data enters a difficulty filter, which removes samples that are too easy or too hard based on model performance, ultimately producing the Curated Dataset.

**First Stage: Rule-based Filter** For four visual reasoning tasks, the following filters are applied:

- Multiple-choice and true/false questions that are prone to hacking are discarded. (Kimi et al., 2025)

- Answers containing symbols such as "=", "[", "]", "(", ")", and ";" are removed, as the absence of these symbols may cause answer mismatches even if the numeric values are correct.

- Answers longer than 20 characters are discarded to avoid overly complex answers.

The filtering process for visual perception tasks involves additional complexity:

- **Detection:** Following Qwen2.5-VL (Bai et al., 2025), data is converted to relative coordinates. Single-box samples contain one box per category, while multi-box samples retain original annotations. Samples with over 10 boxes per category or boxes exceeding 50% of the image are removed. A 1:2 single-to-multi-box ratio is enforced, and category-level long tails are avoided.

- **Grounding:** Data is processed into relative coordinates, and data with a box size greater than 50% of the image is discarded. Complex phrase labels are filtered out.

- **Counting:** Data is balanced per category and only English data is retained.

- **OCR:** Only English OCR data is retained, and final labels must be verifiable by math_verify (Kydlíček, 2025). Since no verifiable reward model (RM) is designed, the OCR task data must pass this validation.

**Second Stage: Difficulty-based Filter**   To remove low-value samples, easy questions already solvable by the base model are filtered out.

For reasoning tasks, we use `Qwen2.5-VL-32B-0321` to compute pass@8, retaining only samples with $0 \leq$ pass@8 $< 100\%$. For perception tasks, specifically detection and grounding, pass@16 is computed using `Qwen2.5-VL-7B` with a 0.5 IoU threshold, and samples with cumulative IoU rewards between 2 and 10 are selected.

All curated data is stored in Parquet format (Apache Software Foundation, 2025) and uniformly mixed for training without online filtering or curriculum scheduling.

## B   INSIGHTS FROM SOURCE-LEVEL MONITORING

While V-Triune provides a powerful framework for unified training, its complexity necessitates robust diagnostics. Our Source-Level Metric Monitoring system proved essential in this regard, allowing us to identify and address several challenges related to both training stability and system scalability. Key issues diagnosed by our system include: (1) catastrophic performance collapse on perception tasks driven by ViT gradient explosion; (2) the leaked generation of `image_token` in responses, particularly for perception tasks; (3) high performance variance on reasoning tasks due to prompt sensitivity; and (4) system memory bottlenecks hindering large-scale evaluation. Informed by these diagnostics, this section details the targeted adjustments we implemented—including freezing the ViT, filtering special tokens, creating a prompt pool, and batch-processing online evaluations—to achieve robust and scalable training.

### B.1   STABILIZING TRAINING BY FREEZING ViT PARAMETERS

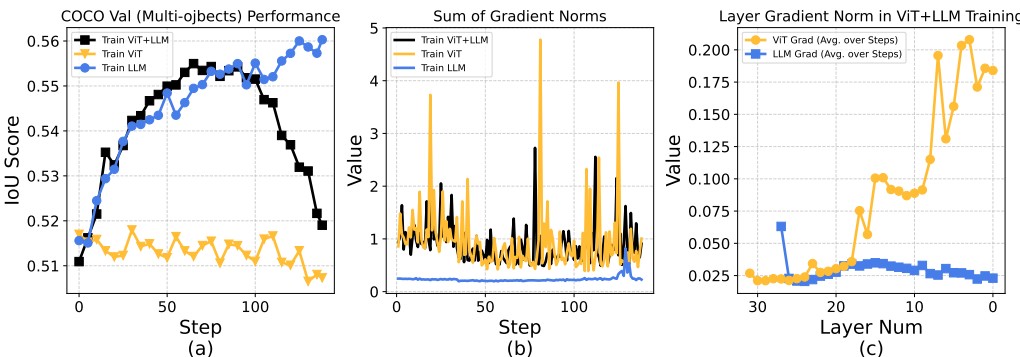

Figure 5: **Analysis of ViT Training Instability.** (a) COCO testset (OOD) performance comparison. (b) Sum of gradient norms under different training schemes. (c) Layer-wise gradient norms of ViT and LLM during full parameter training. Notably, incorporating ViT training leads to a performance decline and highly unstable gradients; remarkably, ViT's gradients amplify during back-propagation, contrasting with the stable layer-wise gradients of the LLM.

In initial experiments, we performed full-parameter training by jointly optimizing the ViT and LLM. However, detection performance consistently collapsed after several dozen steps, regardless of hyperparameter settings. Log analysis revealed unusually large and spiking gradient norms (often ¿1), suggesting instability originating from the ViT. To verify this, we conducted three training configurations: (1) LLM-only, (2) ViT-only, and (3) full-parameter training, all using identical RL settings on Orsta-7B with mixed task data. We monitored: (a) COCO test set performance, (b) total gradient norm, and (c) layer-wise gradient trends during full-parameter training.

As shown in Fig. 5 a, joint training leads to a performance drop, whereas LLM-only training maintains stable gains. ViT-only training yields minimal improvement, indicating that RL benefits primarily stem from updating the LLM. Fig. 5 b shows that ViT training produces significantly higher gradient norms—over 10× larger than LLM-only training.

Layer-wise analysis (Fig. 5 c) confirms this: LLM gradients remain stable across layers, while ViT gradients amplify during backpropagation, with the first layer exhibiting 5–10× larger norms than the last. This gradient explosion destabilizes training and undermines visual performance. Consequently, we freeze ViT parameters in subsequent experiments.

The root cause of this instability remains an open research question, but we offer two key insights. First, RL not only activates VLM capabilities but also enforces modality alignment by grounding responses in visual content. When ViT and LLM are trained jointly, the visual representation—i.e., the alignment target—shifts constantly, leading to instability analogous to the concept drift problem in machine learning (Gama et al., 2014). This dynamic target undermines stable optimization and may cause model collapse. Alternating training, similar to GANs (Goodfellow et al., 2020), where one component is frozen while the other is updated, could offer a solution. Second, ViT's contrastive

pretraining may limit its suitability for RL, as it encourages static, instance-level features rather than the dynamic and causal representations needed for RL tasks. To mitigate this mismatch, auxiliary self-supervised objectives could be introduced during RL to help ViT adapt to the evolving task demands.

## B.2 MITIGATING LEAKED IMAGE SPECIAL TOKENS

To enable accurate advantage estimation, logits for both the query and the generated response are recomputed, as those returned by the inference engine may be imprecise. During the forward pass, image placeholders (highlighted in the red box in Fig. 6, appearing before the "vision_end" token) are replaced with visual features extracted by the ViT and adapter modules. However, the model may mistakenly generate special tokens (highlighted in the blue box in Fig. 6), such as image or video placeholders, that lack corresponding features—particularly under RL-zero settings. To ensure input–feature alignment and maintain training stability, a filtering step is applied to remove all such special tokens from the rollout sequence prior to recomputation.

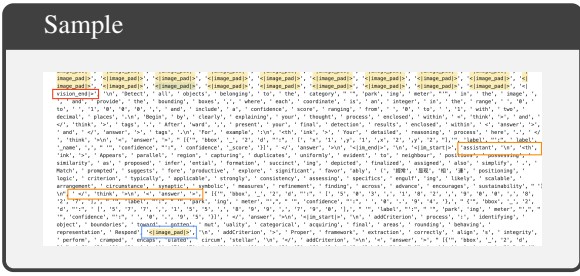

Figure 6: **An Example of Leaked Image Tokens.**

## B.3 MITIGATING PROMPT SENSITIVITY VIA A CoT PROMPT POOL

In the early stages of training for visual mathematics tasks, variations in CoT prompts, despite conveying identical meanings, which can influence model performance, affecting metrics such as accuracy and response length. To reduce this variability, we construct a CoT prompt pool comprising 10 alternatives for "Let's think step by step" and 10 for "Place the answer in \boxed{}." During training, one sentence from each group is randomly selected and appended to the instruction. This strategy mitigates prompt-induced variance and is applied specifically to samples verified with MathVerifyVerifier.

## B.4 ENABLING SCALABLE ONLINE EVALUATION UNDER MEMORY CONSTRAINTS

V-Trinue is implemented atop Verl, a single-controller training framework that can approach system memory limits on the master node, particularly with large-scale vision datasets. To enable effective OOD performance monitoring, we introduce online test-set benchmarking at regular intervals. To mitigate the resulting system overhead, we decouple the testing phase from the main training loop and batch-process benchmarks, bypassing the default vLLM data handling.

## C TRAINING DYNAMICS ANALYSIS

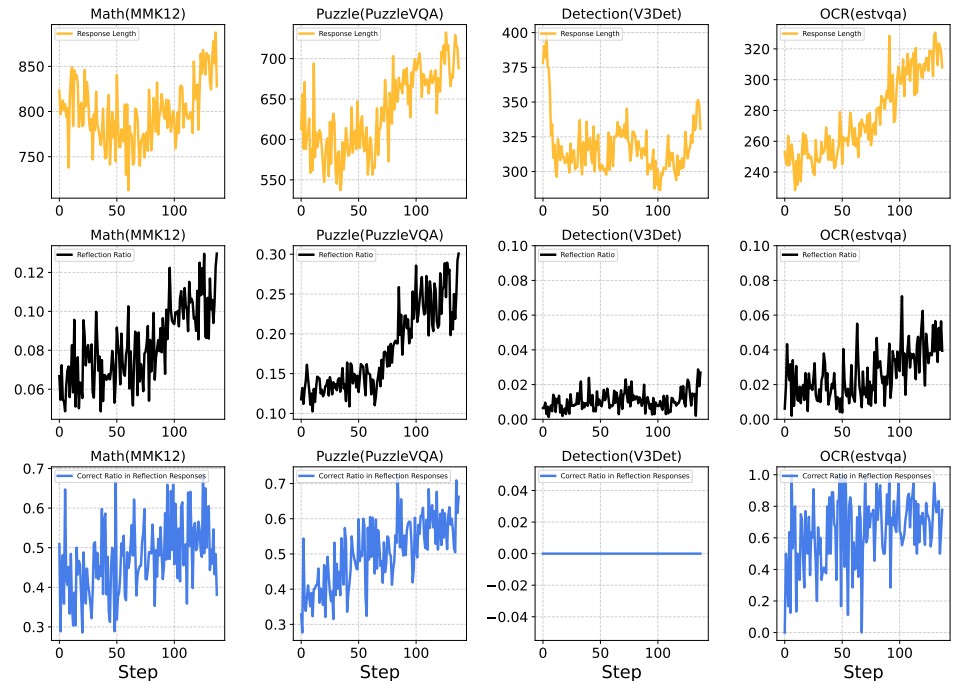

Figure 7: Training dynamics of response length (top row), reflection ratio (middle row), and correct ratio in reflection responses (bottom row) during training steps for Math (MMK12), Puzzle (PuzzleVQA), Detection (V3Det), and OCR (estvqa) tasks using the Orsta-32B-0321 off-policy setting. Each column corresponds to a different task, and each row represents a distinct metric.

V-Triune's source-level metric monitoring enables detailed analysis of cognitive behaviors learned during training, such as reflection patterns. To explore whether perception tasks can benefit from test-time scaling via extended Chain-of-Thought (CoT), we analyze Orsta-32B-0321 off-policy logs. The analysis focuses on four tasks—Math (MMK12), Puzzle (PuzzleVQA), Detection (V3Det), and OCR (estvqa)—each represented by a single dataset. We report three key metrics: response length, reflection ratio, and accuracy of reflection responses, as defined in Sec. 3.3.

As shown in Fig. 7, the evolution of these metrics reveals sharp contrasts across tasks. For reasoning tasks (Math and Puzzle), both response length and reflection ratio consistently increase, indicating a growing reliance on longer, more reflective reasoning. In perception, a clear divergence emerges: OCR's response length and reflection ratio also trend upward, mirroring the reasoning tasks, while Detection's response length decreases before stabilizing, and its reflection ratio remains near zero.

This divergence is further confirmed by reflection quality. The correctness of reflection responses steadily improves for Puzzle and OCR, rising to 0.7–0.8, and fluctuates for Math, indicating partial effectiveness. For Detection, however, it remains at zero throughout, signifying a complete absence of functional reflection.

Overall, reasoning tasks and OCR not only demonstrate increasing reflection usage but also, to varying degrees, improved reflection quality. Detection diverges completely, showing minimal reflective behavior and no benefit from longer, reasoning-style responses. This implies that naively applying extended CoT scaling to all perception tasks may be ineffective; task-specific characteristics, such as those distinguishing OCR from Detection, are critical determinants of a model's learned cognitive strategy.

# D  BENCHMARK DETAILS

To comprehensively assess the model's capabilities, we evaluate performance across three domains: real-world tasks, visual reasoning, and visual perception.

For real-world task evaluation, we employ the core subset of **MEGA-Bench** (Chen et al., 2024), which consists of 440 diverse, long-tailed tasks encompassing over 6,000 expert-curated samples. All results are reported using the official MEGA-Bench evaluation code to ensure consistency.

To evaluate reasoning and knowledge capabilities, we adopt **MMMU** (Yue et al., 2024) and **Math-Vista** (Lu et al., 2023). Both benchmarks are assessed using VLMEvalKit (Duan et al., 2024). Since GPT-4o is used for both answer extraction and scoring—which may introduce variability, we report both rule-based and GPT-4o-based scores for MMMU.

For visual perception evaluation, we include **COCO** (val-2017) (Lin et al., 2014), **OVDEval** (Yao et al., 2023), **CountBench** (Paiss et al., 2023), **OCRBench** (v2) (Fu et al., 2024), and **ScreenSpot-Pro** (Li et al., 2025). We evaluate these benchmarks following their established protocols: Count-Bench is assessed using the VLMEvalKit; OCRBench is evaluated with LMMsEval (Zhang et al., 2024) on the entire dataset; and ScreenSpot-Pro is tested with its official evaluation code. For the detection-focused benchmarks, COCO and OVDEval, we utilize their official codebases to report mAP|map@50 and NMS-mAP respectively. However, for these mAP-based metrics, we adapt the standard calculation protocol. Grounded in the belief that VLM-generated confidence scores are unreliable, our methodology both dismisses scores from the output format (Fig. 8) and employs the model's generation order as a more robust, per-sample ranking criterion (see Appx. E for details). All bounding boxes and keypoints are represented using coordinate values relative to the original input image dimensions.

# E EVALUATION ON COCO

We conduct our evaluation on the COCO val-2017 dataset (Lin et al., 2014), which contains 4,952 images with 36,781 ground-truth bounding boxes. The dataset includes 593 images with a single object (593 boxes) and 4,359 images with multiple objects (36,188 boxes). For the experiment, we use the instruction shown in Fig. 8 to prompt the model to generate a list of all target detections for each of the 4,952 images. The model operates at a temperature of 0 and outputs all bboxes in the format: [{'bbox_2d': [x1,y1,x2,y2],'label': label_name} ...] at one time.

The model's output boxes are parsed into the COCO format, and we use the official cocoapi to calculate the mean Average Precision (mAP). The mAP computation requires a confidence score for each prediction to rank them. We use the predicted box's area relative to the total image area as a pseudo-confidence score. The score is calculated as follows:

$$\text{score} = \frac{(x_2 - x_1) \times (y_2 - y_1)}{\text{image\_width} \times \text{image\_height}}$$

To validate the robustness of our evaluation, we also conducted an ablation study on the choice of the pseudo-confidence function. We implemented and compared several alternative heuristics, including methods based on object position (center_bias) and shape (aspect_ratio), alongside fixed and random baselines. As shown in Tab. 5, the mAP scores are remarkably stable across all deterministic heuristics, with a total spread of around 0.5 mAP. This stability suggests that our evaluation results are not sensitive to the specific choice of the ranking method. Therefore, we adopt the simple and interpretable area_ratio method for all main experiments reported in this paper.

Table 5: Ablation study on pseudo-confidence scoring methods for Qwen2.5-VL-7B-Instruct on the full COCO val-2017 dataset.

| Scoring Method | mAP@50:95 |
|---|---|
| area_ratio (our choice) | 33.63 |
| center_bias | 33.60 |
| aspect_ratio | 33.08 |
| fixed (1.0) | 33.07 |
| random (baseline) | 33.05 |

## F  QUERY EXAMPLE OF DETECTION AND GROUNDING

---
**Query Example of Detection and Grounding**

```
Please detect all instances of the following category within
↪  the image:
{LABEL}.

Let's think step by step and output the final answer in
↪  <answer> and </answer> tags.
For example:
Your detailed reasoning process here.
<answer>
[{'bbox_2d': [x1,y1,x2,y2],'label': label_name}]
</answer>
```
---

Figure 8: **Example query format for detection and grounding tasks.** The query instructs VLMs to identify instances of a given object and format the output in a specific reasoning-answer format.

## G  SAMPLE-LEVEL DATA SCHEME FOR UNIFIED TRAINING

---
**Data Format**

```
{
    "data_source": Value(dtype="string"),
    "images": Sequence(feature=Image(mode=None, decode=True)),
    "prompt": [
        {
            "content": Value(dtype="string"),
            "role": Value(dtype="string")
        }
    ],
    "ability": Value(dtype="string"),
    "reward_model": {
        "answer": Value(dtype="string"),
        "ground_truth": Value(dtype="string"),
        "accuracy_ratio": Value(dtype="float32"),
        "format_ratio": Value(dtype="float32"),
        "verifier": Value(dtype="string"),
        "verifier_parm": Value(dtype="dict")
    },
    "extra_info": {
        "id": Value(dtype="string"),
        "image_path": Value(dtype="string")
    }
}
```
---

Figure 9: **Sample-level Data Scheme for Unified Training**.  This format, implemented using HuggingFace datasets, allows fine-grained control over reward computation by defining `reward_model` (including reward types, weights like `accuracy/format_ratio`) and `verifier` specifications at the individual sample level. This enables flexible and scalable handling of diverse multimodal tasks.

## H DETAILED EXPLANATION OF REFLECTION METRICS

This appendix provides a detailed breakdown of the reflection metrics used in our source-level metric monitoring. These metrics are designed to quantitatively assess the model's self-correction and reasoning processes.

**Reflective Word Set**  Following Ma et al. (2025b), we track a curated list of 15 English words and phrases that indicate a reflective or self-correcting thought process. A response is considered "reflective" if it contains one or more of the following terms:

- re-check, re-evaluate, re-examine, re-think
- recheck, reevaluate, reexamine, rethink
- reevaluation
- check again, think again, try again
- verify, wait, yet

**Metric Definitions**  Based on this word set, we define two metrics:

1. **Reflection Ratio** ($R_{\textbf{reflect}}$): This metric measures the overall frequency of reflective responses. It is defined as the total number of responses containing at least one reflective word ($N_{\text{reflect}}$) divided by the total number of all responses ($N_{\text{total}}$).

$$R_{\text{reflect}} = \frac{N_{\text{reflect}}}{N_{\text{total}}} \tag{4}$$

2. **Correctness Rate within Reflection** ($C_{\textbf{reflect}}$): This metric assesses the effectiveness of the model's reflective reasoning. It is defined as the number of reflective responses that are also correct ($N_{\text{correct\_reflect}}$) divided by the total number of reflective responses ($N_{\text{reflect}}$).

$$C_{\text{reflect}} = \frac{N_{\text{correct\_reflect}}}{N_{\text{reflect}}} \tag{5}$$

We emphasize that this keyword-based approach serves as a rough proxy for reflective behavior, not a precise measurement. It is intended for lightweight, online monitoring to gauge general trends in the model's reasoning process, rather than for a formal or rigorous evaluation of its reflection capabilities.

# I EXPERIMENT ON ADAPTIVE IOU THRESHOLD SCHEDULING.

**Experimental Setup:** To rigorously evaluate the efficacy of adaptive scheduling (proposed by Reviewer wrru) compared to our proposed heuristic curriculum, we implemented a performance-driven controller that dynamically adjusts the IoU threshold based on the model's real-time competence. The models were trained exclusively on the detection and grounding components of our dataset. We monitored the IoU@50 performance of OVDEval (Negation subset) throughout the training process.

We defined a discrete set of progressive IoU milestones $\mathcal{S} = \{0.5, 0.55, 0.6, \ldots, 0.95, 0.99\}$, with training initializing at $T_0 = 0.5$. At each training step $t$, we compute the **Batch Success Rate (BSR)**, defined as the proportion of samples in the current batch where the predicted IoU exceeds the current threshold $T_t$. The threshold for the next step $T_{t+1}$ is updated according to a pre-defined `Target Success` hyperparameter ($\tau$):

$$T_{t+1} = \begin{cases} \text{next}(T_t, \mathcal{S}) & \text{if BSR} > \tau \quad \text{(Promote)} \\ \text{prev}(T_t, \mathcal{S}) & \text{if BSR} < \tau \quad \text{(Demote)} \\ T_t & \text{otherwise} \end{cases} \quad (6)$$

We conducted a ablation study by varying $\tau \in \{0.1, 0.3, 0.5, 0.7, 0.9\}$, covering behaviors ranging from aggressive advancement (low target) to conservative consolidation (high target). The result is shown in Fig. 10.

**Results & Analysis:** 1) **Validation of "Comfort Zone" ($\tau = 0.9$, Red Line):** While initially stable, Fig. 10 c shows the threshold plateaus at 0.85, never enforcing strict supervision (IoU $\geq 0.99$). This lack of high-precision pressure leads to slight performance degradation in later steps (Fig. 10 a). 2) **Instability & Slow Learning ($\tau \leq 0.7$, Other Lines):** Lower targets cause the threshold to spike to 0.99 within 15 steps. This premature difficulty jump introduces severe reward sparsity and oscillation (Fig. 10 b), resulting in slower convergence compared to the $\tau = 0.9$ baseline.

**Conclusion:** These experiments highlight the practical trade-offs in iou threshold scheduling strategies. While the adaptive method may offer theoretical flexibility, it introduces a new hyperparameter (target success $\tau$) that is often less intuitive to calibrate than the IoU threshold itself. Finding a $\tau$ that avoids both stagnation (comfort zone) and reward sparsity is non-trivial. Given these factors, our heuristic dynamic schedule ($0.85 \rightarrow 0.95 \rightarrow 0.99$) is not an arbitrary choice, but a transparent and controllable baseline, ensuring the model reliably transitions to high-precision supervision without the variance associated with performance-driven updates.

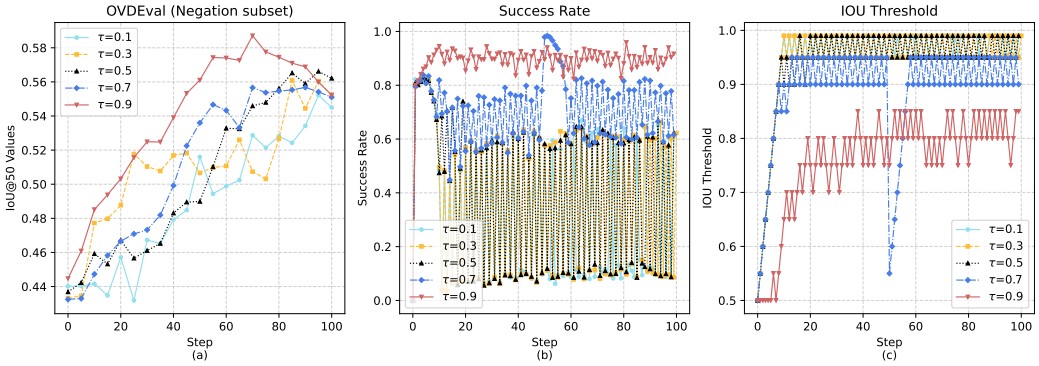

Figure 10: raining Dynamics of Adaptive IoU Scheduling. We vary the target success rate $\tau \in \{0.1, \ldots, 0.9\}$. (a) Validation performance (IoU@50) on OVDEval. (b) Batch Success Rate (BSR) stability. (c) Evolution of the IoU threshold. The results highlight the trade-off between threshold stagnation (at high $\tau$) and premature saturation (at low $\tau$).

## J    SENSITIVITY ANALYSIS AND RATIONALE OF DYNAMIC IoU SCHEDULE

**Experimental Setup:** To validate the robustness of our three-stage Dynamic IoU schedule (0.85 → 0.95 → 0.99), we extended the ablation study presented in Fig. 3(e). The models were trained exclusively on the detection and grounding components of our dataset. We monitored the IoU@50 performance of OVDEval (Negation subset) throughout the training process.

**Variants and Analysis:** We compared our approach against two representative variants: a "Loose" schedule (0.7 → 0.8 → 0.85, Var1) and a "Strict" schedule (0.9 → 0.95 → 0.99, Var2). As illustrated in the IoU@50 trajectory in Fig. 11:

- 1. The "Loose" Variant (Var1) exhibits a rapid initial rise due to high reward density but subsequently stagnates and degrades (black line). This confirms that a low final threshold (0.85) fails to resolve reward ambiguity: the model receives maximal rewards for coarse predictions, losing the gradient incentive for fine-grained refinement.

- 2. The "Strict" Variant (Var2) suffers from a cold-start problem (blue line). The sparse reward signal at the start (IoU $\geq 0.9$) impedes effective gradient estimation, resulting in a significantly slower learning curve in the early stages.

- 3. Ours (Yellow Line) achieves the best learning curve. By initializing at 0.85, we ensure a sufficient signal-to-noise ratio to bootstrap learning; by progressively tightening to 0.99, we prevent the stagnation caused by reward ambiguity observed in Var1.

**Rationale for Thresholds & Three-Stage Design** Our schedule is designed to maximize the optimization range. The starting threshold (0.85 in our case) is selected to calibrate the initial difficulty, targeting a moderate initial reward rate (e.g. $\approx$ 0.2 - 0.4). This specific range can leave ample room for RL improvement. Conversely, the final threshold (0.99) is set strictly to eliminate reward ambiguity, preventing the optimization from stalling at coarse localizations. The intermediate stage (0.95) acts as a buffer to smooth the optimization landscape, allowing the model to adapt its precision gradually.

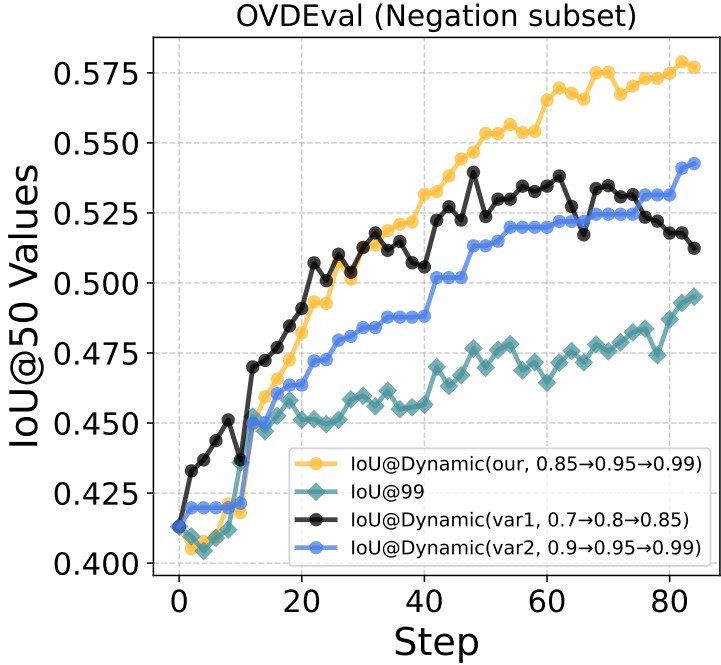

Figure 11: Extended Ablation Study of Fig. 3(e).

