# OpenReview forum: "One RL to See Them All: Visual Triple Unified Reinforcement Learning"
_ICLR.cc/2026/Conference — Submitted to ICLR 2026_

### Official Review · Reviewer_cQWy · 2025-10-26

**Soundness:** 2
**Presentation:** 2
**Contribution:** 2
**Rating:** 4
**Confidence:** 3

**Summary:**

The paper proposes V-Triune, a unified RL system for multimodal models that jointly optimizes visual reasoning and perception within a single pipeline. Implementation-wise, V-Triune uses a sample-wise definition of different rewards (reasoning/perception) that can be used to route into different verifiers.

Using V-Triune, the authors train Orsta (7B, 32B) and report consistent gains (e.g., +2.1 to +14.1 on MEGA-Bench Core) and improvements on several downstream tasks.

A notable technical contribution is the Dynamic IoU curriculum that tightens the IoU threshold during training (0.85 -> 0.95 -> 0.99), balancing reward ambiguity vs. sparsity for detection/grounding.

Overall, I see limited technical contribution in this work and lean to recommend rejection pending author’s discussion.

**Strengths:**

1. Empirical effectiveness is validated through experiments on different scales of models, though it's not clear that the proposed method is the direct cause of the improvement.

**Weaknesses:**

1. Limited technical contribution. In section 3, 3.1 mainly discussed the sample-wise reward dataset format and its corresponding verifier design, 3.2 elaborated a curriculum learning paradigm for IoU-based reward, and 3.3 presented how authors monitor the training process. I think all of them do not provide technical contributions to the current visual RL community.

2. Incomplete ablation studies in the proposed sample-level reward mechanism. In section 4.3, the authors conducted ablations with:

   - Different data mix (reasoning, perception, reasoning + perception)
   - w/ and w/o dynamic IoU annealing strategy

    However, an important ablation is missing – what if we use full data but only switch off the sample-wise reward (i.e. with standard RLVR)? Otherwise there is no way to differentiate the impact of the high-quality training data and the proposed sample-wise reward design.

**Questions:**

1. In L310, the authors mentioned on-/off-policy configuration of their method without explaining. How do you define on-/off-policy in your method? Is your on-policy setup based on a single synchronous loop and the off-policy setup based on a replay buffer?

---

> ### Author Response · Authors · 2025-11-20
> **Response to Reviewer cQWy**
>
> We thank the reviewer for the candid feedback regarding technical contribution and ablation settings.
>
> $\Large \text{W1: Technical Contribution of Section 3 (System vs. Method). }$
>
> The reviewer asks whether Section 3 represents mere engineering or causal methodological contributions. As discussed in General Response 0, V-Triune is a *principled methodology* designed to solve the specific failure modes of unified RL—where naive application fails. The components in §3.1–3.3 are not just formatting or logging, but necessary architectural solutions:
>
> - **§3.1/3.2 (Sample-Level Formatting/Verifier-level rewarding = Contribution 1)**: Solves **Reward Incompatibility**. It uncouples reward logic from the training loop, enabling the joint optimization of heterogeneous signals (exact match vs. IoU) in a single GRPO update. Without such a modular routing abstraction, unified training may still be mathematically possible but practically brittle and hard to scale beyond a few tasks.
>
> - **§3.2.1 (Dynamic IoU = Contribution 2)**: Solves the **Ambiguity-Sparsity Dilemma**. As shown in Fig. 3(c–e); this is a targeted fix for the collapse/stagnation caused by fixed IoU thresholds in detection/grounding tasks.
>
> - **§3.3 (Source-level Monitoring = Contribution 3)**: Solves **System Instability**. It serves as a diagnostic engine that detected and resolved critical failures like ViT gradient explosion and image special token leakage (Appx. B.1 & B.2). It also revealed non-trivial training dynamics such as divergent cognitive strategies across tasks (Appx. C).
>
> $\Large \text{W2: Ablation of the "Sample-level reward mechanism". }$
>
> We clarify that while our schema technically supports arbitrary per-sample reweighting, in this work we only use it to enable heterogeneous batch training via simple, verifier-dependent configurations (specifically: pure accuracy for tasks using `MathVerifyVerifier` , and accuracy + 0.1 format weight for tasks using `DetectionVerifier` ), **with no per-task or per-sample fine-tuning**.
>
> Why we chose standardized settings:
>
> - **Rigorous Attribution**: By deliberately avoiding per-sample fine-tuning, we **isolate the source of improvement**. This design choice helps ensure that the gains reported in §4 primarily reflect the structural advantages of V-Triune, rather than heavy per-task hyperparameter engineering.
>
> - **Lower Bound of Performance**: These results represent a solid baseline. The fact that Orsta achieves SOTA performance (see **GR II**) without granular tuning highlights the framework's inherent effectiveness and potential for further optimization.
>
> The reviewer asks for an ablation “without sample-wise reward.” We agree this is a reasonable sanity check to consider. Conceptually, however, Sample-Level Formatting is routing infrastructure and a system design choice for scalability, rather than a tunable hyperparameter that alters the learning objective:
>
> - **Architectural Necessity**: Without this modular routing, unified training would require **rigid**, non-scalable hard-coding and hard-coding the routing logic (e.g., if task=='math': reward=acc). Crucially, this decoupling abstracts specialized logic, like Dynamic IoU, away from the general training loop, preventing the system from becoming entangled with task-specific schedules.
>
> In future revisions, we are happy to include a sanity check showing that the schema-based and hard-coded implementations indeed match within noise.
>
> $\Large \text{Q: Definition of on-/off-policy in our GRPO setup.}$
>
> We apologize for the ambiguity. Our usage follows standard PPO/GRPO terminology for LLMs:
>
> - "On-policy": Each batch is sampled from the current policy and used for **one** gradient update.
>
> - "Off-policy k": Each batch is sampled once but reused for **k** mini-batch updates (via Importance Sampling) before being discarded. We do not use a replay buffer (like SAC/TD3).

---

> > ### Author Response · Authors · 2025-11-26
> > **Kindly Reminder to Reviewer cQWy**
> >
> > Dear Reviewer cQWy:
> >
> > We have posted detailed responses to address your comments. We would be grateful if you could take a moment to review them. Please let us know if there are any further questions we can clarify.

---

> ### Comment · Reviewer_cQWy · 2025-11-26
>
> Thanks for those clarifications! I updated the presentation sub-score.
>
> However, due to the absence of an important ablation studies to verify the impact of the proposed designs, I decided to maintain my overall evaluation to this work.

---

> ### Author Response · Authors · 2025-11-29
> **Response to Reviewer cQWy (Part II)**
>
> We appreciate the reviewer's response. We would like to clarify as follows:
>
> $\Large \text{W2 (Update): Ablation of the ``sample-level reward mechanism'' and the role of data quality}$
>
> **(a) Concept clarification: our method already uses standard RLVR.**
>
> In our setting, the “sample-level reward” is just standard RLVR (in §3.2: pure accuracy for tasks using `MathVerifyVerifier`, and accuracy + 0.1 * format weight for tasks using `DetectionVerifier`). We do not introduce any fine-grained per-sample weights or heavily-tuned per-task coefficients; we only use a single, shared configuration per verifier (e.g., accuracy and accuracy + 0.1 × format).
>
> On mixed batches, this routing is part of how RLVR is defined: each sample must pick the appropriate verifier. Thus, “full data + standard RLVR without sample-wise reward” is logically equivalent to our current setup; any RLVR implementation must, implicitly or explicitly, route samples to the correct verifier; our sample level data formatting makes this routing explicit and scalable, but the underlying objective is the same as a hard-coded implementation.
>
> **(b) Isolating data quality: curated vs. random 47.7K.**
>
> To separate data quality from design, we trained a counterfactual baseline to compare against our main model (Orsta-7B, trained on the curated corpus). This baseline (Random-47.7K) uses the **exact same RL recipe** (config, steps, backbone) but differs in data source:
>
> - **Curated-47.7K (Orsta-7B)**: the full curated corpus described in Appx. A / Tab. 4.
> - **Random-47.7K (control)**: a stratified random subset from the raw pool (the “Original count” column in Tab. 4). We ensure that the sample count for every data source exactly matches the post-curation counts of Orsta-7B. This keeps the task/source distribution fixed; the only variable is data quality.
>
> **Results**: Orsta-7B (Curated-47.7K) consistently outperforms the Random-47.7K baseline on most benchmarks, with clear gains on complex reasoning (e.g., MME-Reasoning +2.52, Charix(RQ) +2.00) and high-resolution / fine-grained perception (e.g., HrBench4K +2.87, COCO multi-object mAP +1.03). This suggests that the curation design—filtering out reward-harming formatting noise—provides measurable gains beyond simply “having data.”
>
> | Bench | MMMU | MathVista | MathVision | MME-R | Charxiv(RQ)| HrBench | VStar | COCO (M) | OCRBench | ScreenSpot |
> | :--- | :--- | :--- | :--- | :--- | :--- | :--- | :--- | :--- | :--- | :--- |
> | Orsta-7B-Random | 56.33 | 72.2 | 31.57 | 28.62 | 46.4 | 74.38 | **83.25** | 40.38 | 55.83 | 23.85 |
> | Orsta-7B-Curated | **57.1** | **72.5** | **31.73** | **31.14** | **48.4** | **77.25** | 81.68 | **41.41** | **56.05** | **23.91** |
> | *Gain* | +0.77 | +0.3 | +0.16 | +2.52 | +2 | +2.87 | -1.57 | +1.03 | +0.22 | +0.06 |
>
>
> $\small \text{Note: MME-R: MME-Reasoning; Charxiv(RQ): Charxiv (Reasoning Question); COCO (M): COCO (multiple object).}$
>
> **(c) Architecture Robustness: Random vs. Prior Baselines** Importantly, our framework is already strong even without curation. The Random-47.7K model remains competitive with, and often stronger than, prior multi-task VLM-RL systems on key benchmarks (results from General Response II). For example, on COCO multi-object mAP it scores 40.38 vs. 36.58 (VisionReasoner-7B) and 31.54 (VL-Rethinker-7B), and on Charix(RQ) it reaches 46.40 vs. 44.00 (VL-Rethinker-7B). These results indicate that the unified V-Triune framework (unified training + Dynamic IoU) is effective and robust even on uncurated data, while the curated 47.7K corpus further improves performance.
>
> **Note on review integrity.**
>
> We are aware that there were recent issues affecting the anonymity of some reviewers and ACs. We want to reassure the committee that we have not accessed or used any leaked identifying information; all of our responses are written solely based on the officially provided reviews and comments in the ICLR system. We deeply respect the double-blind review process and thank the reviewers and AC for their time and effort.

---

### Official Review · Reviewer_wrru · 2025-10-28

**Soundness:** 3
**Presentation:** 3
**Contribution:** 3
**Rating:** 4
**Confidence:** 4

**Summary:**

This manuscript presents V-Triune, a unified reinforcement learning (RL) framework for jointly training vision-language models (VLMs) on high-level reasoning and low-level perception tasks. It addresses key challenges—data heterogeneity, incompatible reward schemes (e.g., exact-match vs. spatial metrics), and opaque training dynamics—through a three-tier design: Sample-Level Formatting ,Verifier-Level Rewards and Source-Level Monitoring. Using V-Triune, the authors train the Orsta-7B and Orsta-32B models (based on Qwen2.5-VL), which achieve consistent gains over strong baselines on benchmarks like MEGA-Bench, COCO, and MathVista. The work also highlights key engineering insights, such as freezing the Vision Transformer during RL training to avoid gradient explosion and performance collapse.

**Strengths:**

1.Introduced a unified, co-designed, and extensible framework that addresses the intricate challenge of jointly optimizing a single model for both high-level reasoning and fine-grained perception tasks.

2.A "Dynamic IoU Reward" mechanism is proposed for perception tasks, employing a curriculum-based approach that incrementally tightens the IoU threshold to progressively raise the difficulty of earning rewards as training advances.

3.The efficacy and superiority of the proposed method are comprehensively demonstrated through extensive experiments, achieving performance gains across a diverse benchmarks.

**Weaknesses:**

1.While the manuscript posits a synergistic effect from jointly training perception and reasoning tasks, the results on the ScreenSpot-Pro benchmark are not fully convincing. Specifically, the Orsta-7B model demonstrates only a marginal 1.2% performance gain, which lags substantially behind the 50%-60% performance typically achieved by models of the 7B scale on this benchmark. This significant discrepancy casts considerable doubt on the general applicability of the claim that these two task types are always mutually beneficial.

2.The proposed Dynamic IoU Reward schedule, while effective, appears to be based on a pre-defined, heuristic-driven curriculum. This raises the question of whether an adaptive scheduling strategy could yield better or more robust results. For instance, have the authors considered methods where the IoU threshold adjusts automatically based on the model's real-time performance, such as maintaining a target success rate for reward acquisition?

**Questions:**

Regarding the training data composition, the paper reports that 18 distinct datasets were aggregated. For the sake of clarity and reproducibility, could the authors provide a more granular breakdown? Specifically, we request the sample counts for each of the 18 source datasets, detailing the numbers both prior to and following the data curation pipeline. Furthermore, what is the final proportional balance between the samples allocated to reasoning tasks versus those for perception tasks in the curated training corpus?

---

> ### Author Response · Authors · 2025-11-20
> **Response to Reviewer wrru**
>
> We thank the reviewer for the insightful questions on benchmark selection, reward scheduling, and data composition.
>
> $\Large \text{W1: ScreenSpot-Pro performance and evidence of synergy.}$
>
> We appreciate the opportunity to clarify the role of this benchmark.
>
> - **OOD Stress Test**: ScreenSpot-Pro is Out-Of-Domain for our model. Our training data consists primarily of natural scenes and document scans (e.g., from LLAVA-OV[1], V3Det[2]), which are distributionally distinct from the digital GUI images in ScreenSpot. (§3.1 and Appx. A). In this context, despite the significant domain shift, we observe **positive transfer** (+1.2% among 1.6k samples) rather than catastrophic forgetting, validating that the learned spatial representations are robust.
>
> - **Comparison Context**: The 7B models scoring 50%–60% on the leaderboard (e.g., GTA1-Qwen2.5-VL-7B[3]) are **GUI-specialists** trained on massive in-domain datasets (~1.56M GUI samples). It is unfair to expect a general-purpose model trained on 47.7K natural samples to match them.
>
> - **Synergy Evidence**: Our claim of "1+1 > 2 synergy" is primarily based on the in-domain task composition experiments reported in General Response I, where the unified model consistently outperforms specialists on shared tasks under fair budgets.
>
> $\Large \text{W2: Dynamic IoU Schedule vs. Adaptive Strategies.}$
>
> The reviewer correctly identifies our schedule as a pre-defined curriculum. While adaptive strategies (e.g., maintaining a target success rate) are appealing, we prioritized a deterministic schedule for a specific reason: Precision Enforcement.
>
> - **Potential Risk of Adaptive Methods**: A performance-driven controller might stabilize at a "comfortable" but loose threshold (e.g., IoU 0.6) to maintain high reward density. This would fail to force the model into the high-precision regime (e.g. IoU > 0.95) required for fine-grained localization.
>
> - **Our Design Choice**: As discussed in General Response 0, our goal is to resolve the **Ambiguity-Sparsity Dilemma**. Our three-stage schedule (0.85 → 0.95 → 0.99) is a "pragmatic compromise": it ensures dense signals for bootstrapping early on, while forcefully tightening the requirement later to guarantee high-precision convergence. As shown in Fig. 3(c–e); this schedule successfully prevents the collapse seen with fixed thresholds.
>
> Designing a more complex adaptive curricula is interesting and non-trivial and we left as future work, our priority here was ensuring strict convergence to high-precision regions.
>
> $\Large \text{Q1: Data composition and Reasoning-Perception balance.}$
>
> We have added a detailed breakdown table in Appx. A of the revised manuscript.
>
> **Summary Statistics**:
>
> - Reasoning: 27,133 samples (56.8%) – Math, Puzzle, Science, Chart.
>
> - Perception: 20,633 samples (43.2%) – Detection, Grounding, Counting, OCR.
>
> - Total: 47,766 samples. The mixture is relatively balanced, preventing any single domain from dominating the gradient updates.
>
>
> $\Large \text{Reference}$
>
> [1] Li, Bo, et al. "Llava-onevision: Easy visual task transfer." TMLR2024
>
> [2] Wang, Jiaqi, et al. "V3det: Vast vocabulary visual detection dataset." ICCV2023 Oral
>
> [3] Yang, Yan, et al. "Gta1: Gui test-time scaling agent." arXiv:2507.05791

---

> > ### Comment · Reviewer_wrru · 2025-11-20
> >
> > I acknowledge the details provided regarding the dataset composition; however, substantial concerns remain regarding the first two points:
> >
> > 1.Statistical Significance and Domain Gap: First, a marginal 1.2% improvement on a sample size of 1.6k is statistically insignificant and likely falls within the range of random noise. Furthermore, the manuscript posits a "synergistic effect" between Perception and Reasoning, which naturally establishes a higher expectation for performance gains in perception tasks. While ScreenSpot-Pro is indeed OOD, containing professional operations and high-resolution scenarios, the baseline GUI models are also often trained on general GUI data yet achieve drastically better performance. Given that the authors voluntarily selected ScreenSpot-Pro as a benchmark, dismissing poor performance as merely an OOD issue is unconvincing. Have the authors considered incorporating GUI grounding datasets (e.g., UGround, ShowUI) into the training mix to test the hypothesis? Providing experimental results with such data would significantly strengthen the validity of the claimed synergy.
> >
> > 2.Heuristic vs. Adaptive Scheduling: The assertion that adaptive strategies might stagnate in a "comfort zone" remains purely speculative without empirical evidence. Consequently, the proposed three-stage schedule requires rigorous ablation studies to demonstrate its generalization capability rather than it being a manually tuned heuristic. Specifically, what is the rationale behind selecting three stages rather than two? How were the specific IoU thresholds determined? Sensitivity analyses showing how the model performs with alternative threshold values are necessary to justify these specific hyperparameter choices."

---

> ### Author Response · Authors · 2025-11-23
> **Response to Reviewer wrru-Part II**
>
> We thank the reviewer for the constructive suggestions.
>
> $\Large \text{Q3 Additional Experiments on GUI Synergy}$
>
> We incorporated 3k samples from ShowUI (approx. 6% of the total mix) into our training and conducted a controlled ablation under a fixed compute budget (60 steps).
>
> **Results:**
>
> | | ScreenSpot-Pro(~1.6k) | OCRBenchv2(~10k) |
> | :--- | :--- | :--- |
> | (A) Unified (Original) | 23.91 | 55.87 |
> | (B) Perception Only (Original) | 23.78 | 55.33 |
> | (C) Perception + GUI-3k | 29.85 | 55.96 |
> | (D) Unified + GUI-3k (Ours) | 31.68 | 56.09 |
>
> **Analysis:**
>
> **Bridging the Domain Gap:** Adding about 3k in-domain data (~6%) yielded a  +7.77 point gain (rising from 23.91 to 31.68). This substantial jump confirms our initial hypothesis: the performance limitation observed in the original setting was driven by the OOD nature of the benchmark, rather than an intrinsic deficiency in the V-Triune framework. Once the distribution gap is bridged, the model demonstrates strong adaptability.
>
> **Synergy is Real and Amplified by In-Domain Data:** Most importantly, comparing (C) and (D) reveals that the synergy gap significantly widens when domain knowledge is present.
> - Without GUI data, the gap between (A) Unified and (B) Perception was negligible (+0.13).
> - With GUI data, the Unified model outperforms the Perception-only baseline by +1.83 points (31.68 vs 29.85). This result provides strong evidence against the "random noise" concern. It suggests that Reasoning capabilities act as a multiplier: once the model acquires basic visual recognition of GUI elements (via ShowUI data), the reasoning data helps it better ground complex instructions, leading to superior performance than Perception training alone.
>
> $\Large \text{Q4.1  Ablation on Our Dynamic Iou Threshold}$
>
> To address the concern regarding heuristic choices, we extended the ablation study (originally Figure 3e) to perform a more rigorous sensitivity analysis.
>
> **Experimental Setting:** We trained models using only the detection and grounding subsets of our training data. Crucially, we monitored the performance of OVDEval (Negation subset), tracking the IoU@50 performance to visualize the learning curves.
>
> **Variants:** We compared our proposed schedule ($0.85 \to 0.95 \to 0.99$) against "Loose" ($0.7 \to 0.8  \to 0.85$) and "Strict" ($0.9 \to 0.95 \to 0.99$) variants (see Appx. J).
>
> **Results:** As visualized in Fig. 11, the "Loose" variant rises fast but degrades later (black line) due to reward ambiguity from the loose final threshold. The "Strict" variant suffers from a cold-start problem (blue line), lagging significantly in the early stages.
>
> **Rationale for Thresholds & Three-Stage Design**: Our schedule is designed to maximize the optimization range. The starting threshold (0.85 in our case) is selected to calibrate the initial difficulty, targeting a moderate initial reward rate (e.g. ≈ 0.2 - 0.4). This specific range can leave ample room for RL improvement. Conversely, the final threshold (0.99) is set strictly to eliminate reward ambiguity, preventing the optimization from stalling at coarse localizations. The intermediate stage (0.95) acts as a buffer to smooth the optimization landscape, allowing the model to adapt its precision gradually.
>
> $\Large \text{Q4.2   Experiment for Adaptive Iou Threshold}$
>
> We implemented an adaptive IoU threshold scheduler with the range from 0.5 to 0.99 based on Batch Success Rate: the proportion of samples in the current batch where the predicted IoU exceeds the current threshold. Consistent with previous ablations, we trained on detection/grounding data and monitored OVDEval (Negation). We swept the target success rate $\tau \in \{0.1, \dots, 0.9\}$, revealing critical trade-offs (see Appx. I, Fig. 10):
>
> - **Validation of "Comfort Zone" ($\tau=0.9$, Red Line):** While initially stable, Fig. 10(c) shows the threshold plateaus at ~0.85, never enforcing strict supervision ($\text{IoU} \ge 0.99$). This lack of high-precision pressure leads to slight performance degradation in later steps (Fig. 10(a)).
>
> - **Instability & Slow Learning ($\tau \le 0.7$, Other Lines):** Lower targets cause the threshold to spike to 0.99 within 15 steps. This premature difficulty jump introduces severe reward sparsity and oscillation (Fig. 10(b)), resulting in slower convergence compared to the $\tau=0.9$ baseline.
>
> **Conclusion:**  While the adaptive method may offer theoretical flexibility, it does not eliminate the need for tuning but introduces a new hyperparameter (target success $\tau$) that is often less intuitive to calibrate than the IoU threshold itself. Finding a $\tau$ that avoids both stagnation (comfort zone) and reward sparsity is non-trivial. Given these factors, our heuristic dynamic schedule ($0.85 \to 0.95 \to 0.99$) is not an arbitrary choice, but a transparent and controllable baseline, ensuring the model reliably transitions to high-precision supervision without the variance associated with performance-driven updates.

---

> > ### Author Response · Authors · 2025-11-26
> > **Kindly Reminder to Reviewer wrru**
> >
> > Dear Reviewer wrru:
> >
> > We have posted detailed responses to address your comments. We would be grateful if you could take a moment to review them. Please let us know if there are any further questions we can clarify.

---

> > ### Comment · Reviewer_wrru · 2025-11-26
> >
> > Thanks to the author's reply, most of my concerns have been resolved and I have decided to increase the score to 6

---

### Official Review · Reviewer_Mwfy · 2025-10-30

**Soundness:** 2
**Presentation:** 3
**Contribution:** 2
**Rating:** 4
**Confidence:** 5

**Summary:**

This paper proposes V-Triune, a unified reinforcement learning framework that integrates sample-level formatting, verifier-level modular reward computation, and source-level metric monitoring. Based on this framework, the authors train Orsta (7B/32B) models across eight visual reasoning and perception tasks. The method achieves consistent improvements over the base model Qwen2.5-VL, showing stronger performance on both reasoning-heavy and perception-heavy benchmarks.

**Strengths:**

- The paper is clearly structured and easy to follow.
- Compared to the base model Qwen2.5-VL, Orsta surpasses it comprehensively across both reasoning and perception benchmarks.

**Weaknesses:**

- The related work section lacks discussion of prior multi-task or joint training research, and the paper provides no in-depth analysis of this key challenge, despite it being central to the proposed framework.
- In general, the technical novelty is relatively limited. The dynamic IoU reward is a reasonable practice but not particularly insightful; meanwhile, the proposed “source-level metric monitoring” appears to be more of a system or logging design rather than a methodological contribution (Sec. 3.3).
- The task-composition ablation in this paper does not clearly illustrate the difference between per-task fine-tuning and multi-task unified training. It would be better to break down the overall score into specific categories. I am curious whether, on perception benchmarks, the combination of "perception + reasoning" still outperforms "perception" alone.
- The evaluation against other baselines (e.g., MM-Eureka, VL-Rethinker) mainly relies on MEGA-Bench, with insufficient comparisons to other representative benchmarks (e.g., MMMU, MathVista), which are commonly used for evaluation.

**Questions:**

Why does the result in Figure 3(a) not match the main table? The main table reports a score of 38.3, but in Figure 3(a) it is below 37.5.

---

> ### Author Response · Authors · 2025-11-20
> **Response to Reviewer Mwfy**
>
> We thank the reviewer for the constructive suggestions regarding related work, technical contributions, and baselines.
>
> $\Large \text{W1: Positioning against prior multi-task RL and key challenges.}$
>
> We appreciate the suggestion to sharpen our positioning. As noted in Tab.1 and §2, prior works fall into two distinct lines: Reasoning-centric RL[1-3] and Perception-centric RL[4-6]. For example, MM-Eureka[3] and VL-Rethinker[2] that focus on optimizing reasoning tasks through data replay strategies. VisionReasoner[6] designs specialized reward functions for homogeneous perception tasks.
>
> **V-Triune's Unique Niche**: We bridge these regimes by performing unified RL over **8 heterogeneous tasks** spanning both reasoning and perception. This introduces unique joint-training challenges that prior works did not face:
>
> - Reward Incompatibility: Mixing binary (reasoning) and continuous spatial (perception) rewards.
>
> - Ambiguity-Sparsity: Handling heterogeneous reward scales where naive IoU thresholds lead to collapse.
>
> V-Triune addresses these via Sample-Level Formatting (§3.1) and Verfier-level reward computation (§3.2) to unify diverse rewards and Dynamic IoU (resolving ambiguity), enabling stable joint training where tasks synergize rather than compete (see General Response I). We will explicitly add this comparative analysis to §2.
>
> $\Large \text{W2: Technical novelty (Dynamic IoU, Monitoring)}$
>
> We thank the reviewer for this assessment. As detailed in General Response 0, V-Triune is not a new RL primitive, but a *methodological framework* for stable unified VLM-RL training rather than a new RL algorithm.
>
> - Dynamic IoU (C2) is not merely a parameter schedule but a necessary fix for the **"Ambiguity-Sparsity Dilemma"**, without which detection tasks collapse or stagnate during joint training (Fig. 3c–e).
>
> - Source-Level Monitoring (C3) acts as a **diagnostic engine** beyond just logging, identifying and mitigating system-level failures like ViT gradient explosion (Fig. 5) that are specific to this unified regime.
>
> $\Large \text{W3: Breakdown of performance and Does Unified outperform Perception-only on perception tasks?}$
>
> To address this, we evaluated the specific checkpoints from our Figure 3a ablation on 10 diverse downstream benchmarks (see General Response I).
>
> **1. Clarification on Training Setup.** We interpret  "per-task" as domain-specialized training. The models evaluated in GR I are the exact same runs visualized in Fig. 3a, using identical configurations (in Sec. 4.1) to ensure strict control. The only variable is the data mixture.
>
> **2. Does "Perception + Reasoning" outperform "Perception alone"?**
>
> Yes. We analyzed this under two rigorous budget constraints:
>
> - **Under Fixed Compute**: Despite seeing significantly less data (~1.3 vs. 3 epochs), Unified is highly competitive, winning benchmarks like VStar and OCRBench.
>
> - **Under Fixed Data (~2.5 Epochs)**: Unified **consistently outperforms** the Perception-only baseline on perception tasks:
>
>   - COCO Multi-object: Unified (41.41) > Perception-only (36.48). A gain of +4.93 mAP.
>
>   - VStar: Unified (81.68) > Perception-only (80.10).
>
>   - OCRBenchV2: Unified (56.05) > Perception-only (55.74).
>
> **Conclusion**: Reasoning data enhances perception, consistent with a cross-task regularization effect: perception tasks anchor the model’s reasoning in visual cues, while reasoning tasks improve the shared visual–language representation that detection builds on. This benefit appears at the representation level, detection can still use short, direct outputs at inference time, rather than requiring long chains of thought for perception, which is in line with the “cognitive divergence” we observe in Appx. C.
>
> $\Large \text{W4: Comparisons to MM-Eureka, VL-Rethinker beyond MEGA-Bench.}$
>
> As detailed in Gen Resp. II, we conducted a head-to-head evaluation of Orsta-7B against MM-Eureka-7B[3], VL-Rethinker-7B[2], and VisionReasoner-7B[6] on 10 benchmarks. Key Results (see GR II table):
>
> - **Orsta-7B achieves the best score on 7/10 benchmarks**.
>
> - It outperforms reasoning specialists (VL-Rethinker) on perception tasks (e.g., +8.87 mAP on COCO Multi-object) while remaining highly competitive on reasoning.
>
> - It outperforms perception specialists (VisionReasoner) on reasoning tasks (e.g., +2.8 on MathVista) while winning on perception.
>
> This suggests that our unified framework produces *a stronger general-purpose VLM* than baselines optimized for isolated capability families.
>
> $\Large \text{Q1: Apparent mismatch between Figure 3(a) and Table 2.}$
>
> We apologize for the confusion. The score reported in the main table corresponds to the final Orsta-7B checkpoint shown in Figure 2(a). In contrast, Figure 3(a) is a task-composition ablation where we only plot the first 60 training steps for all three runs, because the Perception-only run reaches ≈3 epochs at that point. We will clarify this in the caption to avoid misunderstanding.

---

> > ### Author Response · Authors · 2025-11-20
> > **Response to Reviewer Mwfy (reference)**
> >
> > $\Large \text{Reference}$
> >
> > [1] Yi Yang, et al. R1-onevision: Advancing generalized multimodal reasoning through cross-modal formalization. ICCV 2025
> >
> > [2] Wang, Haozhe, et al. "Vl-rethinker: Incentivizing self-reflection of vision-language models with reinforcement learning." NeurIPS 2025 Spotlight.
> >
> > [3] Meng, Fanqing, et al. "Mm-eureka: Exploring the frontiers of multimodal reasoning with rule-based reinforcement learning." arXiv preprint arXiv:2503.07365.
> >
> > [4] Haozhan Shen, et al. Vlm-r1: A stable and generalizable r1-style large vision-language model. arXiv:2504.07615
> >
> > [5] En Yu, et al. Perception-r1: Pioneering perception policy with reinforcement learning. NeurIPS 2025
> >
> > [6] Liu, Yuqi, et al. "VisionReasoner: Unified Visual Perception and Reasoning via Reinforcement Learning." arXiv:2505.12081.

---

> > > ### Author Response · Authors · 2025-11-26
> > > **Kindly Reminder to Reviewer Mwfy**
> > >
> > > Dear Reviewer Mwfy:
> > >
> > > We have posted detailed responses to address your comments. We would be grateful if you could take a moment to review them. Please let us know if there are any further questions we can clarify.

---

> > > > ### Comment · Reviewer_Mwfy · 2025-11-28
> > > >
> > > > Thank you for the detailed response. The additional experiments are helpful and make the paper more complete, and some of my earlier concerns are addressed. However, after reading the other reviewers’ comments and going through the paper again, I still feel that the work is largely engineering-focused. Joint RL is an important and practical topic, but the analysis and discussion on the joint training aspect remain limited. Most of the insights come from empirical observations, which are valuable to see but not particularly informative for me in understanding the underlying mechanisms.
> > > >
> > > > Given this, I will keep my current score. Thank you again for the thoughtful response, the contribution of dataset, and the extra effort in the revision.

---

> > > > > ### Author Response · Authors · 2025-11-29
> > > > > **Thanks for the reviewer's response**
> > > > >
> > > > > We thank the reviewer for re-reading the paper and for acknowledging that the additional experiments and released dataset improved the paper’s completeness and addressed some of your earlier concerns.
> > > > >
> > > > > Regarding the “engineering-focused” perspective, we respectfully clarify the intended scope. Our work *does not* claim to be a mechanistic interpretability study. In the nascent field of VLM-RL, establishing empirically stable and reproducible training for heterogeneous reasoning + perception tasks is a prerequisite for deeper theoretical analysis.
> > > > >
> > > > > V-Triune contributes as a principled training architecture and framework: it identifies critical failure modes in unified VLM-RL (e.g., reward incompatibility and the ambiguity–sparsity dilemma) and proposes boundary conditions such as Dynamic IoU and source-level diagnostics that are empirically necessary to resolve them.
> > > > >
> > > > > Fully unpacking the theoretical mechanisms of cross-task synergy is an important long-term challenge that we believe exceeds the scope of a single system paper; our goal here is to provide a stable, well-instrumented foundation and curated 47.7K verifiable dataset on top of which such future studies can be meaningfully pursued.
> > > > >
> > > > > **Note on review integrity**
> > > > >
> > > > > We are aware that there were recent issues affecting the anonymity of some reviewers and ACs. We want to reassure the committee that we have not accessed or used any leaked identifying information; all of our responses are written solely based on the officially provided reviews and comments in the ICLR system. We deeply respect the double-blind review process and thank the reviewers and AC for their time and effort.

---

### Official Review · Reviewer_5Epx · 2025-10-31

**Soundness:** 2
**Presentation:** 3
**Contribution:** 2
**Rating:** 4
**Confidence:** 4

**Summary:**

This paper proposes V-Triune, a unified reinforcement learning framework that trains a vision-language model (VLM) on both visual reasoning and perception tasks within one pipeline. The V-Triune system has three key components: (1) Sample-Level Data Formatting (2) Verifier-Level Reward Computation (3) Source-Level Metric Monitoring.  Instead of using a fixed intersection-over-union threshold to reward predicted bounding boxes, the IoU threshold is gradually raised over the course of training. The Dynamic IoU reward provides adaptive and progressive feedback.
Empirically, the RL training yields consistent improvements in both domains: Orsta achieves substantial gains on the MEGA-Bench Core multi-task benchmarks.

**Strengths:**

# Strengths

1. The design of V-Triune demonstrates thoughtful architecture engineering. The Sample-Level Formatting overall is a good implementation strategy, and Verifier-Level Rewards achieves modularity and scalability. The highlighted  Dynamic IoU mechanism gradually tightening the IoU threshold over training to guide the model from coarse to fine localization, which match some insights from previous vision works.


2. The empirical results on multiple benchmarks show the strengths of the proposed methods.

**Weaknesses:**

# Weaknesses

1. Motivation for RL in Perception Tasks: The core premise—that reinforcement learning is the right tool for perception-heavy tasks such as object detection and grounding—is not entirely convincing.

2. Experiments: Although many baselines are reported, the critical point—why the method works—is not sufficiently supported or justified. For example, why not train directly on the collected dataset? Since compute is spent on training, is the proposed method the best way to use it?

3. One detailed observation is that the training is very resource-intensive (64 GPUs, probably many hours for just 3 epochs). RL fine-tuning here generates 8 candidate sequences per prompt and only then gets a reward. That’s a lot of computation for one gradient step. Please show that the performance boost is worth the heavy computation.

4. The proposed approach degrades inference speed (inference efficiency issue) by introducing reasoning steps. For perception tasks where fast responses are essential, this extra time is a drawback; please quantify end-to-end latency/FPS and discuss mitigation (e.g., a no-reasoning fast path).

**Questions:**

# Questions

Does the work freeze the vision backbone? In line 302, the authors state, “Freeze the ViT to prevent gradient explosion.” Does this mean the authors aim to improve perception by updating the LLM while assuming the vision backbone is not the bottleneck?

---

> ### Author Response · Authors · 2025-11-20
> **Response to Reviewer 5Epx**
>
> We thank the reviewer for the thoughtful questions regarding motivation, computational cost, and efficiency.
>
> $\Large \text{W1: Motivation for applying RL to Perception tasks.}$
>
> We clarify that our goal is not to argue that "RL is inherently better than SFT for perception in isolation," but to demonstrate that **unified VLM-RL (Reasoning + Perception) unlocks capabilities that isolated training cannot achieved**.
>
> - **Empirical Justification**: As shown in General Response I, under fixed data budgets, our Unified RL model matches or outperforms the Perception-only RL model on perception benchmarks (e.g., +4.93 mAP on COCO Multi-object).  This pattern indicates that perception tasks benefit from being trained alongside reasoning tasks.
>
> - **Conceptual Fit**: Perception tasks provide verifiable signals (IoU) that are highly amenable to RL. V-Triune leverages this by unifying verifiable reasoning and perception rewards in a single loop, creating a more capable generalist model.
>
> $\Large \text{W2: Why does the method work? }$
>
> The effectiveness stems from cross-task regularization within a unified, verifiable RL framework.
>
> - **Mechanism (Hypothesis)**:  The robust synergy observed in our original ablation (*Fig. 3a on MEGA-Bench*) and the new ablation on 10 downstream benchmarks (General Response I), strongly suggests a mechanism of cross-task regularization: perception tasks anchor the reasoning in precise visual evidence, while reasoning tasks enforce complex deduction. We will discuss this insight in the revised manuscript.
>
> - **Dataset Nature (why not train directly on the collected dataset?)**: Our 47.7K dataset is outcome-based (Query + Answer, no CoT) but lacks intermediate reasoning traces. *Direct training (i.e., outcome-based supervision)* on such targets is prone to shortcut learning and impair the performance of reasoning tasks. RL is structurally superior here, as it optimizes the final correctness via self-rollout, without needing the intermediate problem-solving process.
>
> $\Large \text{W3: Justification of Compute Resources}$
>
> (1) **Standard Cost**: Our setup (8 candidates/prompt) is standard for GRPO-based RL. Using 64 GPUs reflects our choice for faster wall-clock time, not a minimum requirement; the framework works equally well on fewer GPUs with longer training time. Moreover, we do not add any extra trainable components beyond the base VLM; the only overhead comes from the standard GRPO practice of sampling multiple candidates per prompt.
>
> (2) **High ROI**: The compute is well-spent. As shown in General Response I (Fixed Compute Budget), Unified training outperforms the Perception-only baseline (e.g., COCO: 38.62 vs 37.37) using **the exact same training steps**, indicating superior efficiency per gradient update. This efficient synergy ultimately yields the SOTA-level performance reported in General Response II (beating baselines on 7/10 benchmarks).
>
> $\Large \text{W4: Inference efficiency and reasoning overhead on perception.}$
>
> The reviewer raises a valid concern: does RL force slow, long-chain reasoning onto perception? **Our findings indicate the opposite: the model learns adaptive strategies.**
>
> **Evidence 1 (Cognitive Divergence):**
>
> As shown in Appx. C (Fig. 7), source-level monitoring reveals distinct behaviors:
>
> - Reasoning tasks (Math/Puzzle): Response length and reflection ratio increase significantly.
>
> - Perception tasks (Detection): Response length remains short, and reflection ratio stays near zero.
>
> **Evidence 2 (Efficiency & Fast Path):**
>
> We benchmarked Orsta-7B on COCO val-2017 (~5k images) using a single H200 GPU with vLLM v0.11.0 (greedy decoding).
>
> - **Standard Mode (w/ CoT, use the prompt in Appx. F)**: It runs at **25 FPS** (avg. 208 tokens), finishing in <5 mins.
>
> - **Direct Mode (w/o CoT)**: By removing the "think step-by-step" trigger, the model accelerates to **30 FPS** (avg. 93.5 tokens). Crucially, this direct mode slightly improves performance (+0.77 mAP).
>
> In the main experiments and evaluation, we retain a unified CoT-style prompting template across tasks for simplicity and comparability; the direct mode here serves as a task-specific fast path to address the reviewer’s latency concern. This reflects an adaptive strategy: reasoning-rich training refines the shared visual–language representation, but the model learns to invoke explicit reasoning only when needed. For detection, the internalized grounding capability supports a short, direct inference path without degrading accuracy.
>
> $\Large \text{Q1: Rationale for freezing ViT.}$
>
> This was an empirical necessity, not just a preference. As shown in **Figure 5**, early experiments with unfrozen ViT led to gradient explosion (norms > 10x larger than LLM) and *performance collapse* on COCO. In contrast, LLM-only training remained stable. This suggests that for well-pretrained VLMs, the RL alignment bottleneck may lie in the LLM's interpretation of visual features, not the encoder itself.

---

> > ### Author Response · Authors · 2025-11-26
> > **Kindly Reminder to Reviewer 5Epx**
> >
> > Dear Reviewer 5Epx:
> >
> > We have posted detailed responses to address your comments. We would be grateful if you could take a moment to review them. Please let us know if there are any further questions we can clarify.

---

### Official Review · Reviewer_54ER · 2025-11-01

**Soundness:** 3
**Presentation:** 2
**Contribution:** 2
**Rating:** 4
**Confidence:** 3

**Summary:**

The paper proposes V-Triune, a unified reinforcement learning system for post-training vision-language models on both reasoning (math, science, charts, puzzles) and perception (detection, grounding, OCR, counting) tasks. The system has three components: sample-level data formatting, verifier-level reward computation and source-level metric monitoring. A key technical contribution is a Dynamic IoU reward curriculum that tightens the IoU threshold over training, mitigating reward ambiguity vs. sparsity in detection/grounding. Built on this system, the paper trains Orsta-7B/32B on 47.7K curated samples from 18 sources across 8 tasks.

**Strengths:**

1. Unified RL system design with clean separation of concerns.
2. Dynamic IoU curriculum addresses reward ambiguity vs sparsity; backed by ablations on COCO multi-object and OVDEval negation subsets.
3. Clear training/eval configs; data schema; anonymized code/checkpoints promised in supplementary, and provide a 47k dataset to the community.

**Weaknesses:**

1. The core contributions skew toward system integration and scaling—per-sample routing, modular verifiers, and an IoU-based curriculum are each incremental/known ideas; the work reads more like a strong engineering recipe than a new algorithmic principle.
2. The proposed solution involves a lot of moving parts and task-specific logic, which could raise concerns about its scalability and maintainability. Each new task requires custom handling (formatting rules for inputs/outputs and a specialized verifier to compute rewards). For the eight tasks in the paper, the authors defined multiple reward components, e.g. exact answer checks for QA, format compliance checks, IoU calculation for detection, etc., sometimes even with sample-specific weightings. This heavy reliance on manual configuration means that scaling V-Triune to “all” vision-language tasks would demand significant human effort. If one wanted to incorporate a new modality or task (say, segmentation masks or video question answering), they would likely need to design new verifiers or reward functions from scratch. The paper frames V-Triune as extensible, but it does not demonstrate an automatic or learned way to extend to novel tasks, it is not a plug-and-play solution for arbitrary tasks without additional engineering.

**Questions:**

1. How are the various tasks balanced in the unified RL training loop? For example, If a particular task’s reward signal is weaker or noisier, how did you prevent it from being overshadowed by others?
2. How does the RL-fine-tuning approach compare to a multi-task sft on the same collection of tasks? For example, did you train a version of orsta using conventional supervised learning on the eight tasks, and if so, how did its performance differ from the RL-trained orsta?

---

> ### Author Response · Authors · 2025-11-20
>
> We thank the reviewer for the constructive feedback. Below, we address the concerns regarding technical contribution, scalability, and baselines.
>
> $\Large \text{W1: ``Strong engineering framework'' rather than new algorithmic principle.}$
>
> We appreciate this candid assessment. As discussed comprehensively in Gen Resp. 0, V-Triune is indeed a methodological framework designed to solve the specific failure modes of unified RL—where naive application of existing algorithms fails due to reward incompatibility and sparsity.
>
> To briefly reiterate (see GR0 for details):
>
> - Dynamic IoU (C2) is not just a curriculum, but a necessary fix for the **"Ambiguity-Sparsity Dilemma"** in joint training, preventing the collapse observed with fixed thresholds (Fig. 3c–e).
>
> - Source-Level Monitoring (C3) acts as a **diagnostic engine**, identifying system-level failures, such as ViT gradient explosion and image special token leakage that otherwise crash unified training (Fig. 5 & 6) and reveals granular training dynamics, such as the divergent reflection patterns across reasoning and perception tasks (Appx. C).
>
> $\Large \text{W2: Concerns on scalability and "manual configuration" (e.g., Segmentation). }$
>
> The reviewer correctly observes that V-Triune is not "zero-engineering." However, our design principle is **modularity, not repetition**. The `Verifier` (§3.2) acts as a template combining reusable components (e.g., exact match, IoU), enforced via a registry. Taking the reviewer's example of **segmentation**: since mask-IoU is conceptually similar to box-IoU, a developer does not need to start from scratch. They can inherit from `DetectionVerifier` in §3.2 and simply override the accuracy calculation.
>
> Pseudo-code for a Segmentation Extension:
>
> ```python
> @Verifier.register(name="segmentation")
> class SegmentationVerifier(DetectionVerifier):
>     def accuracy(self, predict_str: str, solution: str):
>         # Reuses framework; only replaces IoU logic
>         pred_mask = extract_masks(predict_str)
>         gt_mask = extract_masks(solution)
>         return compute_mask_iou(pred_mask, gt_mask)
> ```
>
> The core RL training loop remains untouched. Human effort is strictly limited to defining the verification rule, which is inherent to the task definition itself, not the learning algorithm.
>
> Once registered, the developer simply sets the sample's verifier key to `"segmentation"` (§3.1). The core GRPO loop remains unchanged. This architecture ensures V-Triune remains maintainable and extensible to new verifiable tasks.
>
> $\Large \text{Q1: How to balance tasks and handle weak/noisy rewards?}$
>
> We do not rely on a single global balancing coefficient. Instead, we ensure balance through:
>
> - **Robust Reward Design**: For detection/grounding, where fixed thresholds inevitably create **noisy (ambiguous)** or weak **(sparse) signals**, the Dynamic IoU schedule (§3.2.1) resolves this dilemma, ensuring stable improvement rather than collapse (Fig. 3d).
>
> - **Diagnostic Monitoring**: We track per-source metrics to check for overshadowing. *Monitoring shows consistent reward growth across all tasks, indicating that no single task dominates the optimization*. Furthermore, as shown in **Appx. C (Fig. 7)**, distinct cognitive strategies emerge: reasoning tasks develop longer, reflective chains, while detection maintains short, precise outputs. This **cognitive divergence** shows that tasks are following their own reward signals rather than being dragged by others.
>
> - **Empirical Synergy**: As shown in General Response I, unified training outperforms specialized training, indicating that tasks mutually benefit each other rather than compete destructively.
>
> $\Large \text{Q2: Comparison to Multi-task SFT Baseline.}$
>
> (1) **Improvements over SFT**: We acknowledge the challenge of constructing an apple-to-apple SFT baseline. Crucially, all Orsta variants are initialized from **Qwen2.5-VL-Instruct**, which is already a state-of-the-art multi-task SFT model. Therefore, the gains reported in §4 are strictly **post-SFT alignment gains**, demonstrating that V-Triune unlocks capabilities beyond what the strong SFT backbone has already achieved.
>
>  (2) **Why no direct SFT baseline**: Our 47.7K dataset is constructed for **outcome-based RL** (Query + Verifiable Answer), lacking the intermediate CoT traces required for high-quality SFT of reasoning tasks. Comparing to a separate SFT run on our small 47k dataset would be redundant (as the backbone has seen millions) and likely inferior due to the lack of CoT.
>
> (3) **Unified vs. Specialized**: To address the spirit of this question (is unified training beneficial?), we conducted a rigorous ablation in Gen Resp. I. Under fixed compute/data budgets, the *Unified RL model matches or outperforms Reasoning-only and Perception-only RL models on their respective benchmarks*. These findings provide strong evidence that V-Triune unlocks genuine synergy  beyond what is achievable by isolated training.

---

> > ### Author Response · Authors · 2025-11-23
> > **Response to W2(Update): Empirical Evidence of Scalability (New GUI Experiment)**
> >
> > To validate this extensibility in practice, we integrated a completely new domain: GUI Grounding (using ShowUI data) into the V-Triune pipeline during the discussion with the Reviewer wrru.
> >
> > This process required zero code modification to the core system or reward logic; we simply formatted the data and routed it to the existing DetectionVerifier. The model successfully learned the task (achieving a +7.77 score increase on ScreenSpot-Pro) without any task-specific engineering.
> >
> > This confirms that for tasks sharing output structures (e.g., bounding boxes / texts), adding new domains is essentially effortless, significantly reducing the marginal cost of scaling.

---

> > > ### Author Response · Authors · 2025-11-26
> > > **Kindly Reminder to Reviewer 54ER**
> > >
> > > Dear Reviewer 54ER:
> > >
> > > We have posted detailed responses to address your comments. We would be grateful if you could take a moment to review them. Please let us know if there are any further questions we can clarify.

---

### Author Response · Authors · 2025-11-20
**General Response 0: What is technically new in V-Triune, and why it matters?**

We thank the reviewers for their candid feedback. While V-Triune leverages standard optimization algorithms (e.g., GRPO), its core contribution is a principled **training framework** that resolves concrete failure modes in unified reasoning + perception RL—a regime where naive application of existing methods often leads to instability.

$\Large \text{1. The Core Challenge: Why Unified VLM-RL is Hard} $

Unified training over 8 heterogeneous tasks is not merely "stacking" datasets. It introduces systematic conflicts that have not been addressed in a single pipeline:

- 1. **Reward incompatibility**. Exact-match rewards (reasoning) and continuous IoU metrics (perception) create conflicting optimization targets in one loop.

- 2. **Ambiguity–sparsity dilemma.** For detection, fixed IoU thresholds that are too loose give many coarse boxes identical reward, while very strict thresholds make rewards extremely sparse, causing collapse or stagnation.

- 3. **System Instability**: Joint training triggers opaque dynamics like ViT gradient explosions and image special token leakage in model responses.

$\Large \text{2. Technical Contributions: An Architecture for Stability and Synergy}$

V-Triune provides three methodological components to address these issues:

- 1. **C1: Sample-Level Formatting & Verifier-Level Reward (§3.1–3.2).** This decouples reward logic from the RL loop and provides three functions: (i) routing — each sample declares its verifier, so heterogeneous rewards (0–1 accuracy, IoU, format) are handled in a single GRPO run; (ii) future-proofing — the schema supports richer reward configurations without changing the trainer;  (iii) diagnostics—the data_source field enables granular monitoring.  Together these can keep the system extensible to new verifiable tasks without rewriting the core RL loop.

- 2. **C2: Dynamic IoU Reward (§3.2.1).** A targeted fix for the ambiguity–sparsity dilemma. Ablations (Fig. 3c–e) show that fixed thresholds result in instability or inefficiency in detection/grounding under unified training. Our dynamic schedule progressively tightens thresholds: early stages provide dense signal to bootstrap learning, later stages enforce high-IoU precision, **enabling stable fine-grained perception gains alongside reasoning**.

- 3. **C3: Source-Level Metric Monitoring (§3.3).** Beyond logging, this serves as a **diagnostic engine**. It detected non-obvious at-scale failures such as ViT gradient explosion and image special token leakage (Appx. B) , directly informing the critical design choice to freeze the ViT and sanitize outputs. Beyond stability, it reveals granular training dynamics, such as the divergent reflection patterns across reasoning and perception tasks (Appx. C).

$\Large \text{3. Empirical Evidence: The Framework Drives the Gains}$

New controlled experiments (detailed in General Response I & II) isolate the impact of our method:

- **1. “Effect of unification under fair budgets.** Under fixed compute/data budgets, the unified model consistently matches or outperforms reasoning-only and perception-only baselines (Gen. Resp. I), indicating that gains come from the unified regime rather than simply “more training”.

- **2. Comparison to strong multi-task RL-VLM baselines.** Orsta-7B outperforms MM-Eureka, VL-Rethinker, and VisionReasoner on 7/10 diverse benchmarks (Gen. Resp. II), validating the effectiveness of our specific framework.

In summary, V-Triune is not just system integration, but a methodological bridge that makes unified, verifiable VLM-RL stable, extensible, and synergistic in practice.

---

> ### Author Response · Authors · 2025-11-20
> **General Response I:  On the empirical effect of unification (synergy vs. compromise)**
>
> We thank Reviewer 54ER for raising the fundamental question:
>
> *Does unified (Reasoning + Perception) RL trade off capabilities, or can it unlock genuine synergy?*
>
> To answer this, we extended our task-composition ablation (originally Fig. 3a on MEGA-Bench) under two fair budgets on 10 downstream benchmarks (the first 5 are reasoning,the last 5 are perception) :
>
> (I) **Fixed Compute:** Same training steps.
>
> (II) **Fixed Data:** Same number of epochs.
>
> $\Large \text{Experimental Setup}$
>
> We evaluated the specific checkpoints from our Figure 3a ablation. To ensure rigorous comparability, all runs use the identical GRPO configuration described in §4.1 (off-policy 8, bsz = 1024, lr=5e-7). **The only variable is the data mixture:**
>
> - (A) Unified (Ours): all 8 tasks (47.7K samples)
>
> - (B) Reasoning-only: 4 reasoning tasks (27.1K samples)
>
> - (C) Perception-only: 4 perception tasks (20.6K samples)
>
> $\Large \text{Table (I) Fixed Training Steps (Compute Budget)}$
>
> Constraint:
>
> (A): 60 step, ≈1.3 epochs
>
> (B): 60 step, ≈2.3 epoch
>
> (C): 60 step, ≈3 epoch
>
> | Ablation | MMMU | M-Vista| Mathvision | MME-R | Charxiv(RQ) | HrBench4K | Vstar | COCO(S\|M) | OCRBenchV2 | ScreenSpotPro |
> | :--- | :---: | :---: | :---: | :---: | :---: | :---: | :---: | :---: | :---: | :---: |
> | (A)Unified | 56.56 | 71.40 | 30.51 | **28.20** | **44.50** | 73.75 | **82.20** | 80.69 \| **38.62** | **55.87** | 23.91 |
> | (B)Reason | 54.89 | **71.70** | **31.24** | 28.03 | 44.00 | 73.25 | 78.53 | 78.36 \| 33.87 | 55.77 | **23.97** |
> | (C)Perception | **56.67** | 68.20 | 29.59 | 26.26 | 41.00 | **76.00** | 80.10 | **80.78** \| 37.37 | 55.33 | 23.78 |
>
> Note: MME-R:  MME-Reasoning; Charxiv(RQ): Charxiv (Reasoning Question); COCO(S|M): COCO (single | multiple object), For COCO, we report both single-object and multi-object mAP; when counting benchmarks, we treat COCO as a single perception benchmark and focus on the more challenging multi-object setting.
>
> $\Large \text{(II) Fixed Training Epochs (Data Budget)}$
>
> Constraint:
>
> (A): 115 step, ≈ 2.5 epochs (aka. Orsta-7B in the main paper)
>
> (B): 65 step, ≈ 2.5 epoch
>
> (C): 50 step, ≈ 2.5 epoch
>
> | Ablation | MMMU | Mathvista | Mathvision | MME-R | Charxiv(RQ) | HrBench4K | Vstar | COCO(S\|M) | OCRBenchV2 | ScreenSpotPro |
> | :--- | :---: | :---: | :---: | :---: | :---: | :---: | :---: | :---: | :---: | :---: |
> | (A)Unified | **57.10** | **72.50** | **31.73** | **31.14** | **48.40** | **77.25** | **81.68** | **80.73** \| **41.41** | **56.05** | **23.91**  |
> | (B)Reason | 56.78 | 70.20 | 30.68 | 28.6 | 43.40 | 72.75 | 79.58 | 78.13 \| 34.90 | 55.59 | 23.72  |
> | (C)Perception | 56.67 | 68.40 | 29.33 | 27.44 | 43.50 | 75.50 | 80.10 | **80.73** \| 36.48 | 55.74 | 23.66  |
>
> $\Large \text{Finding}$:
>
> **Even under a strict fixed-compute budget**, the Unified model matches or outperforms the specialized baselines on 5/10 benchmarks, despite having seen roughly 50% fewer task-specific samples. This superior data efficiency suggests that the observed gains cannot be attributed to longer training alone and are consistent with a mild form of cross-task synergy. An effect that, in our setting, loosely corresponds to a “1+1 > 2” pattern rather than a simple trade-off.
>
>
> $\Large \text{Mechanism: A plausible explanation for the observed trends}$
>
> One plausible explanation for these trends is a form of cross-task regularization. During training, perception tasks anchor the model’s reasoning in precise visual evidence (reducing hallucination), while reasoning tasks refine the shared visual–language representation and instruction-following behavior of the LLM.
>
> Importantly, this synergy manifests at the level of the underlying representation rather than as uniformly longer chains of thought at inference time. In practice, the model learns when explicit reasoning is helpful (e.g., complex math or multi-step puzzles) and when a short, direct response is sufficient (e.g., standard detection), which is consistent with the “cognitive divergence” patterns we observe in Appx. C.
>
> $\Large \text{Conclusion}$
>
> V-Triune does not trade off reasoning and perception in a zero-sum way. Under fair budgets, unified VLM RL training acts as a "win-win": it yields models that are stronger than specialized ones on their home benchmarks, benefiting from the synergy between the two capability families.

---

> ### Author Response · Authors · 2025-11-20
> **General Response II: Comprehensive Baseline Comparison (Orsta-7B vs. SOTA RL-VLMs)**
>
> We thank Reviewer Mwfy for the constructive suggestion to broaden comparisons. To address this, we conducted a comprehensive evaluation of **Orsta-7B** against three strong, contemporary multi-task RL-VLMs: **MM-Eureka-7B** [1], **VL-Rethinker-7B** [2], and **VisionReasoner-7B** [3].
>
> $\Large \text{Experimental Setup}$
>
> We evaluate all models on the **10 diverse benchmarks** introduced in General Response I (5 Reasoning, 5 Perception). To ensure fairness, we used the same eval setting for all models:
>
> - Inference: Greedy decoding (temp=0), max_new_tokens=2048.
>
> - Tools: VLMEvalKit (with GPT-4o judge) for five reasoning benchmarks, Lmms-eval for OCRBenchv2; Official codebases for COCO/ScreenSpot.
>
> $\Large \text{Results:}$
>
> Orsta-7B achieves the **best score on 7/10 benchmarks (bolded)**.
>
> | Model | MMMU | Mathvista | Mathvision | MME-R | Charxiv(RQ) | HrBench4K | Vstar | COCO(S\|M) | OCRBenchV2 | ScreenSpotPro |
> | :--- | :--- | :---: | :---: | :---: | :---: | :---: | :---: | :---: | :---: | :---: |
> | Orsta-7B |  **57.10** | 72.50 | 31.73 | **31.14** | **48.40** | **77.25** | **81.68** | **80.73** \| **41.41** | **56.05** | 23.91 |
> | MM-Eureka-7B |  55.33 | 74.10 (73.0) | 30.84 | 28.45 | 42.10 | 59.62 | 57.07 | 79.73 \| 35.84 | 53.38 | 24.23 |
> | VL-Rethinker-7B |  56.70 (56.70) | **75.40(74.9)** | **32.46 (32.3)** | 29.38 | 44.00 | 65.12 | 68.60 | 72.50 \| 31.54 | 55.70 | **24.48** |
> | VisionReasoner-7B | 56.56 | 69.70 | 29.20 | 25.84 | 41.20 | 74.38 | 80.63 | 80.22 \| 36.58 | 55.44 | 24.23 |
>
> Note: The numbers in parentheses represent the performance reported in the original paper.
>
> $\Large \text{Analysis:}$
>
> This head-to-head comparison reveals two key strengths of V-Triune:
>
> - **Synergy in Reasoning (Wins 3/5)** On reasoning-heavy tasks, Orsta-7B demonstrates robust capabilities, achieving the highest scores on MMMU, MME-Reasoning, and Charxiv(RQ). While VL-Rethinker leads slightly on MathVista/MathVision, Orsta remains highly competitive (within ~1-3 points) while significantly outperforming it on perception tasks.
>
> - **Dominance in Perception (Wins 4/5)** On perception-intensive tasks, Orsta-7B attains the highest score on 4 out of 5 benchmarks. The margin is particularly notable on the challenging **COCO Multi-object** task, where Orsta-7B (41.41 mAP) outperforms the next-best peer (VisionReasoner) by **+4.83 mAP**. This validates the effectiveness of our Dynamic IoU Reward in handling fine-grained localization.
>
> $\Large \text{Conclusion}$
>
> Unlike baselines that often specialize in either reasoning (MM-Eureka/VL-Rethinker) or perception (VisionReasoner), Orsta-7B delivers *SOTA-level performance* across both domains. These results suggest that our unified training framework successfully leverages cross-task knowledge to achieve a robust, general-purpose alignment.
>
> $\Large \text{Reference}$
>
> [1] Meng, Fanqing, et al. "Mm-eureka: Exploring the frontiers of multimodal reasoning with rule-based reinforcement learning." arXiv preprint arXiv:2503.07365.
>
> [2] Wang, Haozhe, et al. "Vl-rethinker: Incentivizing self-reflection of vision-language models with reinforcement learning." NeurIPS 2025 Spotlight.
>
> [3] Liu, Yuqi, et al. "VisionReasoner: Unified Visual Perception and Reasoning via Reinforcement Learning." arXiv preprint arXiv:2505.12081.

---

### Author Response · Authors · 2025-11-30
**Final Author Summary: Reviewer Concerns and New Experiments**

We thank the AC and reviewers for their constructive feedback. After we added the new experiments (e.g., the extended 10-benchmark ablation, GUI grounding with ShowUI, and Dynamic IoU sensitivity/adaptive scheduling), one reviewer (wrru) explicitly raised their overall score to 6.

Several reviewers also highlighted the strengths of our submission, including the clear unified RL system design, the extensible verifier architecture, and the strong empirical performance across both reasoning and perception benchmarks.


Below we summarize, per reviewer, the main concerns and what we added in the rebuttal (linking to the General Responses I & II, denoted GR I / GR II, and the revised sections):


| Reviewer | Main concerns (very brief) | What we did in rebuttal (new experiments / analyses) |
|---------|----------------------------|------------------------------------------------------|
| **54ER** | Novelty vs “engineering recipe”; scalability of verifiers; task balance / SFT. |  Extended the original task-composition ablation (Fig. 3a) to a **10-benchmark study** under fixed-compute and fixed-data budgets in GR I, comparing unified vs reasoning-only vs perception-only RL;  clarified that all Orsta models start from a strong multi-task **SFT backbone**;  and illustrated extensibility both by showing how new tasks (e.g., segmentation) can be added by registering a new verifier without changing the GRPO loop, and by later **integrating a new GUI grounding task (ShowUI) into the same pipeline without modifying the core RL code**. |
| **5Epx** | Motivation for RL on perception; compute cost; inference latency; freezing ViT. | Clarified that we do not claim “RL is always better than SFT for perception in isolation”, but that RL is a natural fit for our **verifiable perception tasks** (IoU-based rewards); in GR I we used the extended 10-benchmark study to show that **perception benchmarks also improve under unified RL vs perception-only and the SFT backbone**; profiled end-to-end detection inference with CoT-style vs fast, no-CoT prompts to show there is no large latency penalty; and used source-level diagnostics (Fig. 5) to justify freezing the ViT due to gradient explosion when it is updated. |
| **Mwfy** | Positioning vs prior multi-task RL; Dynamic IoU / monitoring novelty; “perception+reasoning” vs “perception-only”; broader baselines. | Sharpened related work (§2) to position V-Triune between **reasoning-centric** and **perception-centric** VLM-RL as a unified 8-task regime; used GR I’s extended ablation to explicitly compare **perception+reasoning vs perception-only** on perception benchmarks; and added GR II’s **10-benchmark head-to-head comparison** against MM-Eureka-7B, VL-Rethinker-7B, and VisionReasoner-7B beyond MEGA-Bench. |
| **wrru** | Small gain on ScreenSpot-Pro vs synergy claim; heuristic IoU schedule; data breakdown. | Added a new **GUI grounding experiment** by mixing in ShowUI data to directly test synergy on ScreenSpot-Pro; ran **Dynamic IoU sensitivity studies** and implemented an **adaptive IoU scheduler** as baselines to our 0.85→0.95→0.99 schedule; and provided a full **18-source data-composition table** (reasoning vs perception) in Appx. A. |
| **cQWy** | Section 3 seen as engineering; missing ablation on sample-level reward; on-/off-policy definition. | Clarified that our “sample-level reward” is **standard RLVR made explicit** via per-verifier configs, not per-sample tuning; trained a **Random-47.7K vs Curated-47.7K** control pair under the same RL recipe to isolate the effect of reward-aware data curation; and gave a precise description of our **on-/off-policy GRPO** setup in §4.1 / rebuttal. |

Reviewers generally acknowledged that these additional experiments and clarifications improved the completeness and presentation of the work. **We also confirm that we have not accessed any leaked reviewer or AC identities; all responses and experiments are based solely on the official reviews and comments.**


*We understand that this year’s ICLR review process has been especially challenging, and we are sincerely grateful that you took the time to read our lengthy rebuttal and additional experiments*. We remain confident that the unified VLM-RL framework and evidence we present can meaningfully contribute to multimodal VLM-RL scaling.

---

### Meta-Review · Area_Chair_tQx2 · 2026-01-06

**Summary:**

The paper proposes a unified reinforcement learning system for vision-language models to jointly learn visual reasoning and perception tasks. The system integrates several components like sample-level data formatting and introduces a Dynamic IoU reward to provide progressive feedback. Reviewers acknowledge the engineering quality and empirical gains, but also express concerns (Reviewer 54ER W1, Reviewer 5Epx W1, Reviewer Mwfy W2) about the overall significance of the technical contribution. Although the rebuttal has addressed some issues, these concerns remained influential in my overall assessment.

**Reviewer Concerns:**

**Reviewer 54ER** raised concerns on the significance of the contribution and the scalability of the framework. Similar concerns were also been raised by Reviewer Mwfy.

**Reviewer 5Epx** questioned the motivation and the significance of the proposed method, the costly training and inference speed. The author response regarding inference speed seems convincing, while the rest remains partially outstanding.

**Reviewer Mwfy** raised concerns about insufficient discussion of related works, limited novelty, inadequate empirical comparisons. The reviewer still has major concern on the technical novelty, missing analysis and discussion on the joint training aspect.

**Reviewer wrru** raised concerns about performance gain and heuristic IoU schedule. and increase the score to 6 after author response well addressed their concerns.

**Reviewer cQWy** raised concerns about limited technical contribution and inadequate ablation studies and explicitly express the incline to maintain the score due to the absence of an importance ablation study. The author further provide justification and results. I believe the concerns was mostly addressed.

**Reviewer Scores:**

The discussion between reviewers and authors was unexpectedly terminated early. Area Chairs are therefore asked to put ourselves in the reviewers’ shoes and provide our best estimate of how scores might have changed. While this is challenging, the following reflects only the AC’s best guess:

Reviewer 54ER: 4 → 4\
Reviewer 5Epx: 4 → 4\
Reviewer Mwfy: 4 → 4\
Reviewer wrru: 4 → 6\
Reviewer cQWy: 6 → 6

---

### Decision · Program_Chairs · 2026-01-26

Reject